# On Safety in Safe Bayesian Optimization

**Christian Fiedler**[*]                                              *fiedler@dsme.rwth-aachen.de*
*Institute for Data Science in Mechanical Engineering (DSME)*
*RWTH Aachen University*

**Johanna Menn**[*]                                       *johanna.menn@dsme.rwth-aachen.de*
*Institute for Data Science in Mechanical Engineering (DSME)*
*RWTH Aachen University*

**Lukas Kreisköther**

**Sebastian Trimpe**                                                 *trimpe@dsme.rwth-aachen.de*
*Institute for Data Science in Mechanical Engineering (DSME)*
*RWTH Aachen University*

(* Equal contribution)

**Reviewed on OpenReview:** **https://openreview.net/forum?id=tgFHZMsl1N**

## Abstract

Safe Bayesian Optimization (BO) is increasingly used to optimize an unknown function under safety constraints, a central task in robotics, biomedical engineering, and many other disciplines. Due to the safety-critical nature of these applications, it is crucial that theoretical safety guarantees for these algorithms translate into the real world. In this work, we investigate three safety-related issues in SafeOpt-type algorithms, a popular class of safe BO methods. First, these algorithms critically rely on frequentist uncertainty bounds for Gaussian Process (GP) regression, but concrete implementations typically utilize heuristics that invalidate all safety guarantees. We provide a detailed analysis of this problem and introduce Real-$\beta$-SafeOpt, a variant of the SafeOpt algorithm that leverages recent GP bounds and thus retains all theoretical guarantees. Second, we identify a key technical assumption in SafeOpt-like algorithms, the availability of an upper bound on the reproducing kernel Hilbert space (RKHS) norm of the target function, as a central obstacle to real-world usage. To address this issue, we propose to rely instead on a known Lipschitz and noise bound, and we introduce Lipschitz-only Safe Bayesian Optimization (LoSBO), a SafeOpt-type algorithm using the latter two assumptions. We show empirically that this algorithm is not only safe, but also outperforms the state-of-the-art on several function classes. Third, SafeOpt and derived algorithms rely on a discrete search space, complicating their application to higher-dimensional problems. To broaden the applicability of these algorithms, we introduce Lipschitz-only Safe GP-UCB (LoS-GP-UCB), a LoSBO variant that is applicable to moderately high-dimensional problems, while retaining safety. By analyzing practical safety issues in an important class of safe BO algorithms, and providing ready-to-use algorithms that overcome these issues, this work contributes to bringing safe and reliable machine learning techniques closer to real world applications.

## 1 Introduction

In science, engineering, and business it is often necessary to optimize an unknown target function. Typically, such functions are expensive to evaluate and only noisy function values are available. If it is possible to actively query the function, i.e., to select the inputs that are to be evaluated, this problem is commonly

addressed using Bayesian Optimization (BO), see (Shahriari et al., 2015) or (Garnett, 2023) for an overview. However, in many real-world applications, BO algorithms should avoid the use of certain inputs, frequently for safety reasons. For example, the target function might be a reward function for a robotic task, with the input a control policy for a physical robot. In this example, any inputs leading to physical damage or unsafe behavior should be avoided. An important special case of such a safety constraint is the restriction of query inputs to those with function values not lower than a given threshold (Kim et al., 2020). Inputs that satisfy this constraint are termed safe inputs, and a BO algorithm is considered safe if it queries only safe inputs throughout its run. This type of safety constraint has been introduced in (Sui et al., 2015) and arises for example in biomedical applications (where the target function is patient comfort and the input corresponds to treatment settings) or controller tuning (where the target function is a measure of controller performance and the inputs are tuning parameters). A popular BO algorithm for this problem setting is SafeOpt (Sui et al., 2015). Starting from a given set of safe inputs, this algorithm iteratively searches for a maximum while aiming to avoid unsafe inputs with high probability. It achieves this by utilizing Gaussian Processes (GPs) together with a frequentist uncertainty bound (Srinivas et al., 2010; Chowdhury & Gopalan, 2017) *and* a known upper bound on the Lipschitz constant of the target function. Provided the algorithmic parameters are set correctly, then with (high) pre-defined probability, SafeOpt demonstrably converges to the safely reachable maximum while avoiding unsafe inputs. SafeOpt and its variants have been used in various applications, e.g., safe controller optimization (Berkenkamp et al., 2016), automated deep brain stimulation (Sarikhani et al., 2021) and safe robot learning (Baumann et al., 2021). To ensure safety, SafeOpt and its variants require frequentist uncertainty sets that are valid (i.e., they hold with specified high probability) and explicit (i.e., they can be numerically evaluated). However, two issues arise here: First, the uncertainty bounds from (Srinivas et al., 2010; Chowdhury & Gopalan, 2017) used in most SafeOpt-type algorithms tend to be conservative, even if all necessary ingredients for their computation are available. This restriction can entirely prevent exploration of the target function. Second, such uncertainty bounds rely upon a particular property of the target function (a known finite upper bound on the Reproducing Kernel Hilbert Space (RKHS) norm), which in practice is very difficult to derive from reasonable prior knowledge. As a consequence of these issues, algorithmically usable uncertainty sets are not yet available for SafeOpt. Indeed, to the best of our knowledge, all implementations of SafeOpt and its variants have instead used heuristics (Berkenkamp et al., 2016; Kirschner et al., 2019a; Baumann et al., 2021; Koller et al., 2019; Helwa et al., 2019).[1] This shortcoming means that in practice such implementations lose all their theoretical safety guarantees. In this work, we carefully investigate this issue and propose practical solutions, bringing safe BO closer to real-world usage.

**Outline** In Section 2, we recall some background material. The problem setting and our objectives are presented in Section 3, and Section 4 provides a comprehensive discussion of related work. In Section 5, we investigate safety issues of SafeOpt-type algorithms arising from using heuristics instead of valid frequentist uncertainty bounds. In Section 6, we discuss the practical safety problems arising from the assumption of a known upper bound on the RKHS norm of the target function, and propose to rely instead on a known Lipschitz and noise bound to address this issue, with Lipschitz-only Safe Bayesian Optimization (LoSBO) as an appropriate algorithm for this. To enable this type of safe BO beyond low dimensional input spaces, Section 7 introduces another algorithmic variant, called Lipschitz-only Safe Gaussian Process-Upper Confidence Bound (LoS-GP-UCB), which is suitable for moderately high dimensions. Finally, Section 8 closes the article with a summary and outlook. Additional background, discussion and experimental details can be found in the appendix, and the source code for all experiments is available in a GitHub repository.[2]

## 2 Background

In this section, we provide a brief review of Gaussian Process (GP) regression, reproducing kernel Hilbert spaces (RKHSs), and frequentist uncertainty bounds for GP regression, as these three components form the foundations for SafeOpt-type algorithms.

---

[1] The precise experimental settings are not reported in (Sui et al., 2015). However, based on the descriptions provided, it can be inferred that some form of heuristic was used.

[2] https://github.com/Data-Science-in-Mechanical-Engineering/LoSBO

### 2.1 Gaussian processes and reproducing kernel Hilbert spaces

A GP is a collection of $\mathbb{R}$-valued random variables, here indexed by the set $D$, such that every finite collection of those random variables has a multivariate normal distribution. A GP $g$ is uniquely defined by its mean function $m(x) = \mathbb{E}[g(x)]$ and covariance function $k(x, x') = \mathbb{E}[(g(x) - m(x))((g(x') - m(x')))]$, and we denote such a GP by $g \sim \mathcal{GP}_D(m, k)$. In GP regression, we start with a prior GP. Assuming independent and identically distributed (i.i.d.) additive normal noise, $\epsilon_t \sim \mathcal{N}(0, \sigma^2)$, this prior GP can be updated with data $(x_1, y_1), \ldots, (x_t, y_t)$, leading to a posterior GP. Without loss of generality we assume that the prior GP has zero mean. Then the posterior mean, posterior covariance and posterior variance are given by $\mu_t(x) = \boldsymbol{k}_t(x)^T (\boldsymbol{K}_t + \sigma^2 \boldsymbol{I}_t)^{-1} \boldsymbol{y}_t$, $k_t(x, x') = k(x, x') - \boldsymbol{k}_t(x)^T (\boldsymbol{K}_t + \sigma^2 \boldsymbol{I}_t)^{-1} \boldsymbol{k}_t(x')$, and $\sigma_t^2(x) = k_t(x, x)$, respectively, where $\boldsymbol{y}_t = [y_1, ..., y_t]^T$ is the vector of observed, noisy function values of $f$, the kernel matrix $\boldsymbol{K}_t \in \mathbb{R}^{t \times t}$ has entries $[k(x, x')]_{x, x' \in D_t}$, the vector $\boldsymbol{k}_t(x) = [k(x_1, x) \cdots k(x_t, x)]^T$ contains the covariances between $x$ and the observed data points, and $I_t$ is the $t \times t$ identity matrix. In practice, the prior mean, prior covariance function, and noise level, are (partially) chosen based on prior knowledge. If no specific prior knowledge regarding the mean is available, the zero function is usually chosen. Furthermore, in practice these three components are only partially specified, usually up to some parameters, which are then called hyperparameters in the context. In BO, the latter are often determined during the optimization via hyperparameter optimization (Garnett, 2023).

If $k$ is the covariance function of a GP, then it is symmetric and positive semidefinite.[3] Conversely, if $k$ is the reproducing kernel of a RKHS, then $k$ is symmetric and positive semidefinite, and there exists a GP having $k$ as its covariance function, and the GP can be chosen with a zero mean function (Berlinet & Thomas-Agnan, 2004). Finally, for every symmetric and positive semidefinite function $k$ (and hence for every covariance function of a GP) there exists a unique associated Hilbert space of functions, called a *reproducing kernel Hilbert space* (RKHS) and denoted by $(H_k, \langle \cdot, \cdot \rangle_k)$, with (i) $k(\cdot, x) \in H_k$ for all $x \in D$, and (ii) $f(x) = \langle f, k(\cdot, x) \rangle_k$ for all $f \in H_k$ and $x \in D$. For the reader's convenience, Section A.1 contains a brief introduction and discussion of these function spaces.

For more details on GPs, GP regression, and related method, we refer to (Rasmussen & Williams, 2006) and (Garnett, 2023), and for more background on RKHSs, we refer to (Steinwart & Christmann, 2008, Chapter 4).

### 2.2 Frequentist uncertainty bounds

An important ingredient in SafeOpt-type algorithms are upper and lower bounds on the unknown target function, and these bounds have to hold uniformly in both time and input space. If we adopt a stochastic setup, then this can be formalized by finding upper and lower bounds such that for a given user-specified confidence $\delta \in (0, 1)$, we have $\mathbb{P}[\ell_t(x) \leq f(x) \leq u_t(x), \forall x \in D, t \geq 0] \geq 1 - \delta$, where the probability is with respect to the data-generating process, while $f$ is assumed fixed and nonrandom. In GP regression, the posterior mean $\mu_t$ can be interpreted as a nominal estimate of $f$, and the posterior variance $\sigma_t^2$ as a measure of uncertainty of this estimate. However, using the posterior variance to build upper and lower bounds in the SafeOpt setting is not straightforward. Firstly, the posterior variance is a *pointwise* measure of uncertainty about the ground truth, but the upper and lower bounds have to hold *uniformly* over the input set, and also uniformly in time. Secondly, GP regression is by its nature a Bayesian method. However, the SafeOpt setting is a typical frequentist setup – we have a fixed, yet unknown ground truth, about which we receive noisy information. In particular, any stochasticity arises only through the data-generating process (e.g., via random noise on the function values), and not through epistemic uncertainty, as is the case in the Bayesian setup. This difficulty is well-known, see (Fiedler et al., 2021a), and is particularly relevant in the context of robust control and related areas (Fiedler et al., 2021b). We thus need bounds $(\nu_t)_{t \geq 1}$, such that for a user-specified $\delta \in (0, 1)$ we have

$$\mathbb{P}[|f(x) - \mu_t(x)| \leq \nu_t(x) \, \forall x \in D, t \geq 1] \geq 1 - \delta. \tag{1}$$

The bounding function $\nu_t$ must only depend on data collected up to time $t$, as well as reasonable prior knowledge about $f$ and the data-generating process, e.g., about the noise statistics. These bounds are

---

[3] This means that all matrices $(k(x_i, x_j))_{i,j=1,\ldots,N}$ for $x_1, \ldots, x_N \in D$ and $N \in \mathbb{N}_{>0}$ are symmetric positive semidefinite.

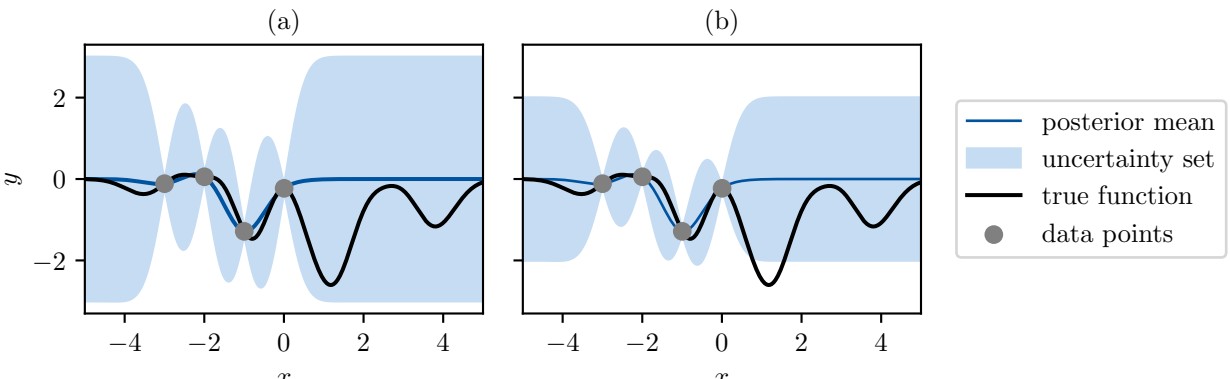

Figure 1: Illustration of the required GP error bounds. Consider a fixed ground truth (solid black line), of which only finitely many samples are known (gray dots). Applying GP regression leads to a posterior GP fully described by the posterior mean (solid blue line) and the posterior variance, from which a high-probability uncertainty set can be derived (shaded blue). Left: The ground truth is completely contained in the uncertainty set. Right: The ground truth violates the uncertainty bound around $x = 1$.

illustrated in Figure 1. All common bounds are of the form

$$\nu_t(x) = \beta_t \sigma_t(x), \tag{2}$$

where $\beta_t \in \mathbb{R}_{\geq 0}$ is some scaling factor. Let $k$ be the covariance function used in GP regression, and $H_k$ the unique RKHS with $k$ as its reproducing kernel. Assume that the ground truth is contained in this RKHS, i.e., $f \in H_k$. Let $\mathbb{F} = (\mathcal{F}_t)_{t \geq 0}$ be a filtration defined on the underlying probability space, and assume that the sequence of inputs $(x_t)_{t \geq 0}$ chosen by the algorithm is adapted to $\mathbb{F}$.[4] The first bound of the form (2) was introduced in the seminal work (Srinivas et al., 2010), cf. their Theorem 6, which holds in the case of bounded noise. This is also the bound that was used in the original SafeOpt paper (Sui et al., 2015). At the moment, the most commonly used uncertainty bound in the analysis of SafeOpt-type algorithms is (Chowdhury & Gopalan, 2017, Theorem 2). Assume that $(\epsilon_t)_t$ is a martingale difference sequence that is conditionally $R$-subgaussian w.r.t. $\mathbb{F}$ for some $R \in \mathbb{R}_{\geq 0}$, which holds for example for i.i.d. bounded or normal noise. Additionally, assume that $\lambda > 1$, or $\lambda \geq 1$ and the covariance function $k$ is positive definite, where $\lambda$ is the nominal noise variance used in GP regression. Under these conditions, the uncertainty bound (1) holds with

$$\beta_t = \|f\|_k + R\sqrt{2(\gamma_{t-1} + 1 + \log(1/\delta))} \tag{3}$$

in (2), where $\gamma_t$ is the *maximum information gain* after $t$ rounds, cf. (Srinivas et al., 2010) for a thorough discussion of this quantity. In contrast to (Srinivas et al., 2010, Theorem 6), this bound allows subgaussian noise (including bounded noise and normal noise) and involves only fairly small numerical constants. However, it still requires the maximum information gain or an upper bound thereof, which can be difficult to work with in practice, and it introduces some conservatism. Motivated by these shortcomings, Fiedler et al. (2021a) proposed a data-dependent scaling factor in (2), based on (Chowdhury & Gopalan, 2017, Theorem 2). Assume the same setting as this latter result, and also assume that the covariance function $k$ is positive definite, then we can set

$$\beta_t = \|f\|_k + \frac{R}{\sqrt{\lambda}}\sqrt{\ln\left(\det(\bar{\lambda}/\lambda\mathbf{K}_t + \bar{\lambda}\mathbf{I}_t)\right) - 2\ln(\delta)}, \tag{4}$$

where we defined $\bar{\lambda} = \max\{1, \lambda\}$, and $\lambda$ is again the nominal noise variance used in GP regression, corresponding to the regularization parameter in kernel ridge regression. This bound no longer involves the maximum information gain, and numerical experiments demonstrate that the resulting uncertainty bounds are rarely significantly larger than common heuristics, see (Fiedler et al., 2021a). In fact, the bounds are

---

[4]Of course, this requires that $D$ is a measurable space, which is not a problem in practice.

sufficiently small to use in algorithms, e.g. (Fiedler et al., 2021b). Finally, from the results in the doctoral thesis (Abbasi-Yadkori, 2013), which was published in 2012, an uncertainty bound can be deduced that is superior to (Chowdhury & Gopalan, 2017, Theorem 2), and therefore also improves over (4). Consider the same setting as introduced above. Combining Theorem 3.11 with Remark 3.13 in (Abbasi-Yadkori, 2013), we find that for all $\delta \in (0,1)$ we can set

$$\beta_t = \|f\|_k + \frac{R}{\sqrt{\lambda}} \sqrt{2 \ln \left( \frac{1}{\delta} \det \left( \mathbf{I}_t + \frac{1}{\lambda} \mathbf{K}_t \right) \right)} \tag{5}$$

in (2). Like (4), this bound can be easily evaluated, but it appears to have been used only infrequently in the machine learning community[5], though it has recently been rediscovered, for example, in (Whitehouse et al., 2024).

## 3 Problem setting and objectives

We now formalize our problem setting following the seminal paper (Sui et al., 2015), describe SafeOpt-type algorithms in detail, and specify our objectives for the remainder of this work. Consider a nonempty set $D$, the *input set*, and a fixed, but unknown function $f : D \to \mathbb{R}$, the *target function* or *ground truth*. We are interested in an algorithm that finds the maximum of $f$ by iteratively querying the function. At time step $t \in \mathbb{N}_0$, such an algorithm chooses an input $x_t \in D$ and receives a noisy function evaluation $y_t = f(x_t) + \epsilon_t$, where $\epsilon_t$ is additive measurement noise. As a safety constraint, all chosen inputs must have a function value above a given safety threshold $h \in \mathbb{R}$, i.e., $f(x_t) \geq h$ for all $t$. Furthermore, the algorithm should be sample-efficient, i.e., use as few function queries as possible to find an input with a high function value. To make progress on this problem, it is clear that some restriction on the function $f$ must be posed. Central to our developments is the next assumption.

**Assumption 1.** *$D$ is equipped with a metric $d : D \times D \to \mathbb{R}_{\geq 0}$. Additionally, $f$ is $L$-Lipschitz continuous, where $L \in \mathbb{R}_{\geq 0}$ is a known Lipschitz constant.*

The second assumption means that for all $x, x' \in D$, we have $|f(x) - f(x')| \leq L\, d(x, x')$. From now on, we work under Assumption 1. Furthermore, we assume that we have access to a non-empty set of known safe inputs $S_0 \subseteq D$, i.e., for all $x \in S_0$ we have $f(x) \geq h$. The SafeOpt algorithm and its derivatives use an iteratively updated model $\mathcal{M}_t$ that provides estimated upper and lower bounds $u_t$ and $\ell_t$ on $f$, i.e., with a certain confidence it holds that $\ell_t(x) \leq f(x) \leq u_t(x)$ for all $x \in D$ and all $t \geq 1$. These bounds are also used to provide a measure of uncertainty defined as $w_t(x) = u_t(x) - \ell_t(x)$. In each step $t \geq 1$, the previous model $\mathcal{M}_{t-1}$ together with the Lipschitz assumption is used to determine a new set $S_t \subseteq D$ of safe inputs, starting from the initial safe set $S_0$. Subsequently, a set $M_t \subseteq S_t$ of potential maximizers of the target function, and a set $G_t \subseteq S_t$ of potential expanders is computed. The latter contains inputs that are likely to lead to new safe inputs upon query. Finally, the target function is queried at the input $x_t = \operatorname{argmax}_{x \in G_t \cup M_t} w_t(x)$, a noisy function value $y_t$ is received, and the model $\mathcal{M}_{t-1}$ is updated with the data point $(x_t, y_t)$. Different variants of SafeOpt result from different choices of models and computations of $S_t$, $M_t$ and $G_t$. To the best of our knowledge, in all SafeOpt-type algorithms, the unknown ground truth $f$ is modeled as a GP. To compute appropriate upper and lower bounds, it is assumed that an appropriate scaling factor $\beta_t$ is available, cf. (2). For each time step $t$, define $C_t(x) = C_{t-1} \cap Q_t(x)$, where $Q_t(x) = [\mu_{t-1}(x) \pm \beta_t\, \sigma_{t-1}(x)]$, and, starting with $Q_0(x) = \mathbb{R}$ for all $x \in D$, $C_0(x) = [h, \infty)$ for all $x \in S_0$ and $C_0(x) = \mathbb{R}$ for $x \in D \setminus S_0$. The corresponding estimated upper and lower bounds are given by $u_t(x) = \max C_t(x)$ and $\ell_t(x) = \min C_t(x)$, respectively. In the original SafeOpt algorithm from (Sui et al., 2015), for each step $t \geq 1$, the new safe sets are calculated by

$$S_t = \bigcup_{x \in S_{t-1}} \{x' \in D \,|\, \ell_t(x) - L\, d(x, x') \geq h\}, \tag{6}$$

the potential maximizers are given by

$$M_t = \{x \in S_t \,|\, u_t(x) \geq \max_{x_S \in S_t} \ell_t(x_S)\}, \tag{7}$$

---

[5]Note however that for $0 < \lambda < 1$ and $k$ positive definite, (5) reduces to (4).

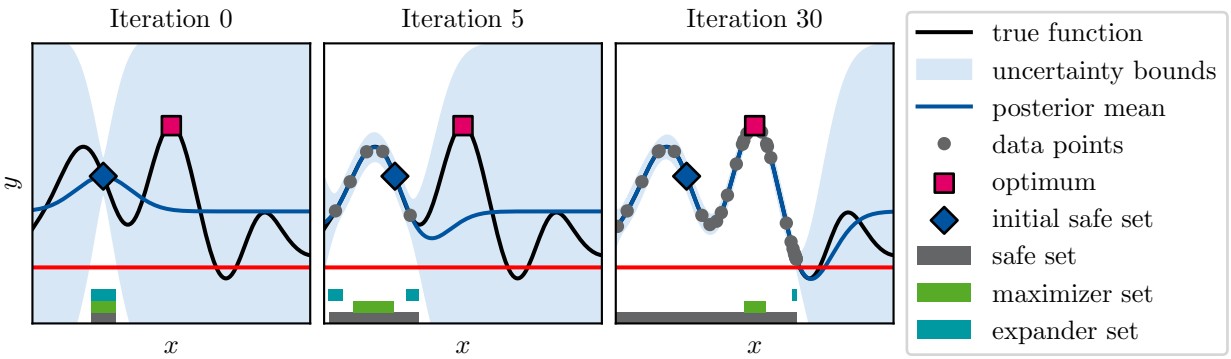

Figure 2: Illustration of SafeOpt. The safe set (gray bar), expanders (blue bar), and maximizers (green bar), derived from the current GP model (with solid blue line the posterior mean, shaded blue areas the uncertainty sets), are used to find the safely reachable optimum (red box). In each iteration, the next input is chosen from the union of the current expanders and maximizers (a subset of the safe set) by maximizing the acquisition function.

---

**Algorithm 1** SafeOpt

**Require:** Lipschitz constant $L$, algorithm to compute $\beta_t$, initial safe set $S_0$, safety threshold $h$

1: $Q_0(x) \leftarrow \mathbb{R}$ for all $x \in D$         $\triangleright$ Initialization of uncertainty sets
2: $C_0(x) \leftarrow [h, \infty)$ for all $x \in S_0$
3: $C_0(x) \leftarrow \mathbb{R}$ for all $x \in D \setminus S_0$
4: **for** $t = 1, 2, \ldots$ **do**
5:     $C_t(x) \leftarrow C_{t-1}(x) \cap Q_{t-1}(x)$ for all $x \in D$    $\triangleright$ Compute upper and lower bounds for current iteration
6:     $\ell_t(x) \leftarrow \min C_t(x),\, u_t(x) \leftarrow \max C_t(x)$ for all $x \in D$
7:     **if** $t > 1$ **then**                                           $\triangleright$ Compute new safe set
8:        $S_t = S_{t-1} \cup \{x \in D \mid \exists x_s \in S_{t-1} : \ell_t(x_s) - Ld(x_s, x) \geq h\}$
9:     **else**
10:        $S_1 = S_0$
11:     **end if**
12:     $G_t \leftarrow \{x \in S_t \mid \exists x' \in D \setminus S_t : u_t(x) - Ld(x, x') \geq h\}$        $\triangleright$ Compute set of potential expanders
13:     $M_t = \{x \in S_t \mid u_t(x) \geq \max_{x_S \in S_t} \ell_t(x_S)\}$          $\triangleright$ Compute set of potential maximizers
14:     $x_t \leftarrow \operatorname{argmax}_{x \in G_t \cup M_t} w_t(x)$                       $\triangleright$ Determine next input
15:     Query function with $x_t$, receive $y_t = f(x_t) + \epsilon_t$
16:     Update GP with new data point $(x_t, y_t)$, resulting in mean $\mu_t$ and $\sigma_t$
17:     Compute updated $\beta_t$
18:     $Q_t(x) = [\mu_t(x) - \beta_t\, \sigma_t(x), \mu_t(x) + \beta_t\, \sigma_t(x)]$ for all $x \in D$
19: **end for**

---

and the potential expanders by

$$G_t = \{x_S \in S_t \mid \exists x \in D \setminus S_t : u_t(x_S) - Ld(x_S, x) \geq h\}. \tag{8}$$

The resulting algorithm is illustrated in Figure 2. A formal description of the algorithm using pseudocode is provided by Algorithm 1. In some popular variants of the SafeOpt algorithm, no Lipschitz bound is used in the computation of the safe sets (Berkenkamp et al., 2016). However, since knowledge of such a Lipschitz bound is additional knowledge that should be used by the algorithm, and we strongly rely on this assumption from Section 6 onwards, we do not consider these algorithmic variants in the present work.

Our primary objective is to investigate and improve practically relevant safety aspects of SafeOpt-type algorithms. We will consider three specific objectives, which will be explored in Sections 5, 6 and 7, respectively.

**(O1)** While SafeOpt and related algorithms come with precise theoretical safety guarantees, all known implementations to date use heuristics (usually by setting $\beta_t$ to some constant) instead of theoretically sound uncertainty bounds. These heuristics generally invalidate all theoretical safety guarantees, raising the question of whether their use leads to safety problems in practice.

**(O2)** Since SafeOpt and its variants are adopted for scenarios with stringent safety requirements where no failure should occur with high probability and no tuning phase is allowed, it is important that all underpinning assumptions are reasonable and verifiable by users. The most delicate of these assumptions is knowledge of an upper bound of the RKHS norm of the target function, raising the question of whether the RKHS norm bound assumption is reasonable, and if not, how it can be replaced.

**(O3)** In many relevant applications of safe BO, the input space is a continuous domain in moderately high dimensions. Since SafeOpt-type algorithms rely on a discrete input space, they cannot be used in these settings. It is therefore necessary to devise alternatives that work in moderately high dimensions, while retaining safety guarantees.

## 4 Related work

In the following, related work will be reviewed. We first provide an overview of safe BO methods, particularly those closely related to the present setting. As Lipschitz bounds play a central role from Section 6 onwards, we also review previous work on methods that rely on such an assumption.

### 4.1 Safety in Bayesian optimization

We first consider safety in the context of Bayesian optimization (BO) (Shahriari et al., 2015; Garnett, 2023). The type of safety constraint considered in this work was first introduced in the seminal paper (Sui et al., 2015), which also proposed and analyzed the original SafeOpt algorithm. Several variations to the original algorithm have been proposed, with a SafeOpt-variant that does not use a Lipschitz constant (Berkenkamp et al., 2016) proving particularly popular in the robotics community. While the original SafeOpt algorithm from Sui et al. (2015) interleaves safe exploration and function optimization over certified safe sets, a two-stage variant was introduced and analyzed in (Sui et al., 2018). Since SafeOpt relies on a discrete input space, and hence requires discretization in the case of a continuous optimization problem, applying this algorithm to continuous problems in even moderate dimensions can be very challenging, cf. also Section 7. This motivated the introduction of a variant based on swarm-optimization (Duivenvoorden et al., 2017), albeit with heuristic safety guarantees, as well as a variant tailored to dynamical problems in high dimensions (Sukhija et al., 2023). A method based on random projections was also proposed and analyzed in (Kirschner et al., 2019a), and applied to tuning of a particle accelerator (Kirschner et al., 2019b). In terms of problem formulations, instead of just one safety constraint for the input function, multiple safety constraints can also be enforced, each encoded by an unknown constraint function (Sui et al., 2018; Berkenkamp et al., 2023). In many applications of SafeOpt-type algorithms, properties of a dynamical system need to be optimized under safety constraints, for example, in controller tuning with BO. The dynamic aspect can be included in the optimization algorithm and its safety mechanisms, for example, in order to use a backup controller (Baumann et al., 2021). Furthermore, while formally SafeOpt-type algorithms are BO algorithms having input constraints, this is different from *constrained BO* as considered for example in (Hernández-Lobato et al., 2016). In the latter type of BO, the focus is on finding good inputs (i.e., corresponding to a high objective function value) that fulfill the (usually unknown) constraints, and violation of the constraints during the optimization process is not considered problematic (though of course it can be advantageous to avoid this). By contrast, safe BO aims to avoid constraint violations during the optimization process since these are considered problematic or even harmful. For an in-depth discussion of this difference, and further connections between the two problem settings, we refer to the excellent survey (Kim et al., 2020). SafeOpt-type algorithms are motivated by problems where safety violations are considered to be very costly, so *any* safety violations should be avoided with very high probability. This contrasts with a whole spectrum of related, but different safe BO (and reinforcement learning) settings. For example, in robot learning a

small amount of crashes might deemed acceptable, in which case a crash-aware BO algorithm (Marco et al., 2021) would be preferable over a SafeOpt-type algorithm. Similarly, in the context of bandit algorithms (Slivkins et al., 2019; Lattimore & Szepesvári, 2020), instead of avoiding a bad option at all costs, one might instead want to be *conservative*, carefully exploring alternatives, starting from a default strategy. This setting is formalized in the context of *conservative bandits* (Wu et al., 2016), and again, is related to, but distinct from, the SafeOpt setting. Corresponding bandit formulations in the context of hard safety are also available (Amani et al., 2019), where the focus is on regret bounds under safety constraints. Finally, parallel to the situation of conservative bandits, there are *cautious* variants of BO (Fröhlich et al., 2021).

## 4.2 Lipschitz-based methods

In order to address safety-related issues uncovered and discussed in Sections 5.1 and 6.1, we will introduce algorithms based on a Lipschitz assumption. While regularity properties like Lipschitz (and the more general Hölder) continuity play an important role in the theory of statistical learning, especially in nonparametric statistical estimation (Tsybakov, 2009), learning algorithms based on Lipschitz assumptions have received relatively little attention in the machine learning community. Relevant exceptions to this are Lipschitz bandits (Kleinberg et al., 2019) and the original SafeOpt algorithm from Sui et al. (2015). The situation is considerably different in the field of global optimization and the systems and control community, respectively. In the former, Lipschitz continuity with a specific Lipschitz constant is a standard assumption, used in a variety of algorithms for (certified) global optimization, cf. (Hansen et al., 1992; Pintér, 1995), though usually a noise-free setting is assumed in this literature. Global optimization of Lipschitz functions has recently also received attention from the machine learning community (Malherbe & Vayatis, 2017). Similarly, a specified Lipschitz constant is also used in the context of Lipschitz interpolation (Beliakov, 2006). Furthermore, closely related to our approach taken in Section 6, a deterministic variant of SafeOpt has been considered in Sergeyev et al. (2020). However, the latter reference only works with functions on a compact interval, and does not use any BO techniques. In the systems and control community, Lipschitz assumptions have long been used, especially in the context of systems identification, where they were explicitly introduced and popularized by Milanese & Novara (2004), though similar methods had been used before, e.g., (Cooper, 1995). A known bound on the Lipschitz constant and on the size of additive noise is commonly used to derive uncertainty bounds in the context of regression. This approach has been further popularized and extended to the case of Hölder continuous functions by Calliess (2014), widely known as *kinky inference* in the systems and control community. A central assumption in this context is knowledge of a *concrete, numerical* upper bound on the Lipschitz constant of the target function. This assumption has a clear geometric and practical interpretation, namely a bounded rate of change of the target quantity. As such, it is related to the well-established field of sensitivity analysis (Da Veiga et al., 2021), for example. Approaches to estimate the Lipschitz constant of an unknown function have been proposed both in the context of global optimization (Strongin, 1973) and in Lipschitz-based regression methods, particularly in the context of systems identification (Milanese & Novara, 2004; Novara et al., 2013; Calliess et al., 2020), see (Huang et al., 2023) for an overview and very recent sample-complexity results. We would like to stress that these approaches are not suitable for the present setting of hard safety constraints, since the estimation of a Lipschitz constant bound requires queries to the target function, which in turn already need to be safe, see also the discussion in Section 6.1. The developments in Sections 6 and 7 combine kernel-based methods (here GP regression) with a Lipschitz assumption to overcome the requirement of a known bound on the RKHS norm of the target function, see Section 6.1 for details. The problematic nature of an RKHS norm bound in the context of learning-based control has been recognized for some time (Lederer et al., 2019; Fiedler et al., 2022). In (Lederer et al., 2019), using probabilistic Lipschitz bounds together with a space discretization was suggested to derive GP uncertainty bounds. However, this approach relies on a probabilistic setting, and is therefore not suitable in the context of SafeOpt-type algorithms. The work (Fiedler et al., 2022) proposes the usage of geometric constraints as prior knowledge in the context of uncertainty sets for kernel-based regression, with Lipschitz constant bounds as a special case. The resulting kernel machines, which provide nominal predictions and smoothed uncertainty bounds adhering to the geometric constraints, are not necessary in our setting, though using more general geometric constraints than Lipschitz constant bounds might be a promising avenue in the context of SafeOpt-type algorithms. Finally, combining kernel methods with Lipschitz assumptions is a natural approach as there is a close connection between regularity properties of a kernel and Lipschitz

continuity of functions in the RKHS generated by the kernel. For a thorough discussion of this point, we refer to (Fiedler, 2023).

# 5 Frequentist uncertainty bounds and practical safety issues in SafeOpt

We now investigate practical safety implications of commonly used heuristics in the frequentist uncertainty bounds in SafeOpt-type algorithms, addressing objective (O1). In Section 5.1, we discuss why these heuristics are problematic and demonstrate safety issues using numerical experiments. To overcome these issues, in Section 5.2, we propose using state-of-the-art frequentist uncertainty bounds in the actual algorithm.

## 5.1 Practical safety issues in SafeOpt

Safety in SafeOpt-type algorithms is ensured by restricting query inputs to safe sets, which are computed using frequentist uncertainty bounds, typically in the form (2) using (3). However, these bounds are often too conservative for algorithmic use, which has led to implementations adopting heuristic choices for $\beta_t$, for example, $\beta_t \equiv 2$ in (Berkenkamp et al., 2016; Turchetta et al., 2016), $\beta_t \equiv 3$ in (Helwa et al., 2019; Baumann et al., 2021), or $\beta_t \equiv 4$ in (Sukhija et al., 2023). Using such heuristics instead of evaluating $\beta_t$ **invalidates all safety guarantees**. In practice, choosing some $\beta_t$ can be a useful heuristic, as demonstrated by the reported success of SafeOpt-type algorithms (Berkenkamp et al., 2016; Baumann et al., 2021; Sukhija et al., 2023). However, it should be stressed that in the setting of SafeOpt as outlined in Section 3, the learning algorithm has to fulfill a **hard safety constraint** – namely, that no unsafe inputs are queried by the algorithm (potentially only with high probability). In particular, no burn-in or tuning phase for $\beta_t$ is allowed. Such heuristics not only invalidate the theoretical safety guarantees, but can also actually lead to safety violations.

First, we demonstrate empirically that a simple heuristic like setting $\beta_t \equiv 2$ can lead to a significant proportion of bound violations. To do so, we follow the general approach adopted in (Fiedler et al., 2021a). We randomly generate 100 RKHS functions on $[0, 1]$ with RKHS norm 10 using the squared exponential (SE) kernel, and for each of the functions, we generate 10000 data sets, run GP regression on it, and finally check whether the uncertainty set (2) with $\beta_t \equiv 2$ fully contains the respective function. 2727±3882 (average ± SD) of these runs (all 10000 repetitions for all 100 functions) led to a bound violation, a sizeable proportion. Second, we show that these bound violations can indeed lead to safety violations when running SafeOpt. We generate an RKHS function $f$ and run SafeOpt with $\beta_t \equiv 2$, but all other algorithmic parameters set correctly (or even conservatively) for 10000 independent runs. This leads to 2862 (out of 10000) runs with safety violations (see Table 1), which is unacceptable for most application scenarios of SafeOpt-type algorithms.

These experiments illustrate that using heuristics in SafeOpt can be highly problematic. Even in the relative benign setting used above, both uncertainty bound and safety violations occur. Moreover, in application scenarios for SafeOpt-type algorithms, tuning the heuristic scaling factor is not possible, as it is the primary mechanism for safety. Dispensing with the need for such heuristics, and retaining safety guarantees both in theory and practice, is therefore the primary motivation for this work.

## 5.2 Real-$\beta$-SafeOpt

As a first step, we propose using modern uncertainty bounds in SafeOpt that can be computed numerically, avoiding the replacement with unreliable heuristics. For this purpose, we investigate the original SafeOpt algorithm, as described by Algorithm 1, with $\beta_t$ from (5). To clearly distinguish this variant of SafeOpt from previous work, we call it *Real-$\beta$-SafeOpt*, emphasizing that we use a *theoretically sound choice of $\beta_t$*. The bound (5) requires the determinant to be computed, which can be computationally expensive, but typical applications of SafeOpt and related algorithms allow few evaluations, so this does not pose a problem. Furthermore, the additive noise needs to be a conditionally $R$-subgaussian (martingale-difference) sequence with a (known upper bound on) $R$. This assumption is standard, has a clear interpretation, and is in many cases harmless. Finally, for a frequentist uncertainty bound, we also need the next assumption.

**Assumption 2.** *Some $B \in \mathbb{R}_{\geq 0}$ is known with $\|f\|_k \leq B$.*

Combining these ingredients allows us to compute the bound (5), so Real-$\beta$-SafeOpt can actually be implemented. We would like to stress that both the underlying algorithm (SafeOpt) and the used uncertainty bound have been known, so we do not claim any new technical developments here. However, to the best of our knowledge, this concrete version of SafeOpt has not been considered in the literature so far, and no empirical evaluation on it has been conducted so far. To illustrate the advantages of Real-$\beta$-SafeOpt, we run it on the same function $f$ as before (cf. Section 5.1), and set $\delta = 0.01$, as well as the true RKHS norm in (5). Running this experiment results in 0 failures (cf. Table 1), evidently well within the $\delta$ range required. An obvious question arising at this point is how Real-$\beta$-SafeOpt compares in terms of performance with previous SafeOpt-variants relying on heuristics. Unfortunately, a meaningful comparison is impossible since the two algorithmic variants address different problems. If for the latter the heuristic constant $\beta$ is determined by trial-and-error, and SafeOpt is then run with this constant that leads to no or almost no safety violation, then we essentially end up with a different algorithm overall. If the heuristic constant $\beta$ is chosen arbitrarily or based on experience with previous usages of SafeOpt-type algorithms, then the original setting of hard safety constraints is abandoned, and instead we end up with a form of cautious BO. By contrast, Real-$\beta$-SafeOpt tries to stay within the original setting of SafeOpt that requires hard safety constraints. To investigate how Real-$\beta$-SafeOpt typically behaves, we will perform a careful empirical evaluation of this algorithm variant in Section 6.4.

# 6 Lipschitz-only safe Bayesian optimization (LoSBO)

In this section, we address objective (O2), in particular, we investigate the central Assumption 2 of a known upper bound on the RKHS norm of the target function, finding that this is a problematic assumption in practice. To overcome this issue, we propose to rely instead on a known Lipschitz and noise bound, and introduce Lipschitz-only Safe Bayesian Optimization (LoSBO) as an appropriate algorithm, on which we perform extensive numerical experiments.

## 6.1 Practical problems with the RKHS norm bound

A central ingredient in the Real-$\beta$-SafeOpt algorithm is an upper bound on the RKHS norm of the target function. In particular, the safety and exploration guarantees inherited from (Sui et al., 2015) hinge on the knowledge of a concrete *numerical upper bound* on the RKHS norm. Unfortunately, while the RKHS norm is very well understood from a theoretical point of view, in practice it is currently not possible to derive concrete numerical upper bounds on the RKHS norm from realistic assumptions in non-trivial cases, to the best of our knowledge, cf. the additional discussion in Section A.2. This issue is known in e.g. the learning-based control community (Fiedler et al., 2022), but has not yet been addressed in the context of safe BO. However, it has immediate consequences for SafeOpt-type algorithms. These algorithms are intended for hard safety settings, i.e., scenarios where *any* safety violation is very costly and must be avoided. In these scenarios it is not possible to have a tuning phase before the actual optimization run where algorithmic parameters are set, since the safety requirements hold from the start. For SafeOpt-type algorithms this means that all algorithmic parameters have to be set beforehand, and these parameters need to ensure the safety and lead to satisfying exploration behavior. However, this entails an upper bound on the RKHS norm of the target function, which currently appears to be impossible to derive from reasonable prior knowledge in practically relevant scenarios.

Furthermore, using an invalid RKHS upper norm bound can indeed easily lead to safety violations. In order to illustrate this, we run Real-$\beta$-SafeOpt on the same function $f$ from Section 5.1, and set $\delta = 0.01$, but this time, we use a misspecified RKHS norm of 2.5 in (5). Running this experiment now results in 1338 failure runs out of 10000, cf. Table 1, which is much more than what would be expected from a safety probability of $1 - \delta = 0.99$. For a summary of this experiment, see again Table 1. Finally, simply using a very conservative upper bound on the RKHS norm is not a viable strategy to overcome this problem. A severe overestimation of the RKHS norm leads to very large and overly conservative uncertainty bounds, which in turn leads to performance issues. In particular, since the uncertainty bounds are used to determine the safety sets in SafeOpt-type algorithms, a supposed RKHS upper norm bound that is too conservative can result in the algorithm "getting stuck", i.e., no more exploration is possible. It should also be noted

that it is not even clear what "very conservative" means in the present context. While an extensive body of theory and strong qualitative intuitions on these objects are available, cf. Section A.2, the lack of concrete, quantitative bounds amenable to numerical evaluation makes it very difficult for users of BO to judge what a conservative estimate of the RKHS norm in a given situation could be.

We argue that as a consequence, any SafeOpt-like algorithm *should not depend on the knowledge of an upper bound on the RKHS norm of the target function for safety*, at least in the setting of hard safety requirements where *any* failure should be avoided (with high probability). More generally, in order to guarantee safety, we should only use assumptions that are judged as reliable and reasonable by practitioners. In particular, all assumptions that are used for safety should have a clear interpretation for practitioners and a clear connection to established prior knowledge in the application area of the safe BO algorithm. In the end, it is up to the user of safe BO to decide which assumptions used in the safety mechanism can be considered as reliable.

## 6.2 Describing LoSBO and its safety guarantees

Motivated by the popularity of SafeOpt, which combines GPs with a Lipschitz assumption, and the extensive experience of the systems and control community with Lipschitz bounds and bounded noise (cf. Section 4), we propose to use *an upper bound on the Lipschitz constant of the target function and a known noise bound* as the ingredients for safety in BO. The key idea is to ensure that the safety mechanism works reliably independent of the (statistical) exploration mechanism. In a generic SafeOpt algorithm, safety is guaranteed by ensuring that the safe sets $S_t$ contain only safe inputs (potentially only up to high probability), i.e., requiring that $f(x) \geq h$ for all $x \in S_t$. Once this property is fulfilled, the rest of the algorithm can no longer violate the safety constraints anymore. Based on the earlier discussion, the construction of the safe set should only rely on the Lipschitz and noise bound. As is well-known, these two assumption allow the construction of lower bounds on the function (Milanese & Novara, 2004), and the corresponding safe sets should therefore be defined for all $t \geq 1$ as

$$S_t = S_{t-1} \cup \{x \in D \mid y_{t-1} - E - Ld(x_{t-1}, x) \geq h\}, \tag{9}$$

where $L \in \mathbb{R}_{\geq 0}$ is a bound on the Lipschitz constant of the unknown target function, $E \in \mathbb{R}_{\geq 0}$ a bound on the magnitude of the noise, and $S_0$ is the initial safe set. We propose using this variant of the safe set, and leaving the rest of the generic SafeOpt algorithm unchanged, which leads to an algorithm we call Lipschitz-only Safe Bayesian Optimization (LoSBO). Our proposed modification applies to *any* algorithm instantiating the generic SafeOpt strategy outlined in Section 3. For concreteness, we focus in the following on the original SafeOpt algorithm from (Sui et al., 2015). The algorithm fulfills the following safety guarantee.

**Proposition 1.** *Let $f : D \to \mathbb{R}$ be an $L$-Lipschitz function. Assume that $|\epsilon_t| \leq E$ for all $t \geq 1$ and let $\emptyset \neq S_0 \subseteq D$ such that $f(x) \geq h$ for all $x \in S_0$. For any choice of the scaling factors $\beta_t > 0$, running the LoSBO algorithm leads to a sequence of only safe inputs, i.e., we have $f(x_t) \geq h$ for all $t \geq 1$.*

*Proof.* It is enough to show that $\forall t \geq 0$ and $x \in S_t$, we have $f(x) \geq h$. Induction on $t$: For $t = 0$, this follows by assumption. For $t \geq 1$, let $x \in S_t = S_{t-1} \cup \{x \in D \mid y_{t-1} - E - Ld(x_{t-1}, x) \geq h\}$. If $x \in S_{t-1}$, then $f(x) \geq h$ follows from the induction hypothesis. Otherwise we have

$$f(x) = f(x_t) + \epsilon_t - \epsilon_t + f(x) - f(x_t) \geq y_t - E - Ld(x_t, x) \geq h,$$

where we used the $L$-Lipschitz continuity of $f$ and the noise bound $|\epsilon_t| \leq E$ in the first inequality, and the definition of $S_t$ in the second inequality. $\qquad\square$

The argument in the proof above is well-known in e.g. the systems identification literature, and the resulting bounds even fulfill certain optimality properties (Milanese & Novara, 2004). We would like to stress that the safety guarantee of LoSBO, as formalized in Proposition 1, is *deterministic*, i.e., it always holds and not only with high probability. This type of safety is often preferred in the context of control and robotics (Hewing et al., 2020; Brunke et al., 2022).

### 6.3 Discussion

While LoSBO arises from a rather minor modification of the generic SafeOpt algorithm class (by changing the computation of the safe sets $S_t$), **on a conceptual level significant differences arise**. Inspecting the proof of Proposition 1 shows that the safety guarantee of LoSBO is *independent* of the underlying model sequence $(\mathcal{M}_t)_t$. As an important consequence, the choice of the uncertainty sets used in the optimization of the acquisition function cannot jeopardize safety. One consequence is that the assumption that the target function $f$ is contained in the RKHS of the covariance function used in GP regression is not necessary anymore. In particular, *in order to ensure safety, we need only Assumption 1* together with a noise bound, *and not Assumption 2 anymore*. Similarly, hyperparameter tuning is not an issue for safety, cf. also to our discussion in Section 2.2. Of course, an appropriate function model is important for good exploration performance, but this issue is now independent of the safety aspect. As another, even more important consequence, in the context of the concrete LoSBO variant described in Algorithm 3 (Appendix), the scaling parameter $\beta_t$ is now a proper tuning parameter. Modifying them, even online, in order to improve exploration no longer interferes with the safety requirements. This differs from previous variants (and practical usage) of SafeOpt, where the scaling factors $\beta_t$ *cannot* be freely tuned since they are central for the safety mechanism of these algorithms. This aspect is illustrated in Figure 3. In the situation depicted there, the uncertainty bounds do not hold uniformly, i.e., the target function is not completely covered by them, and deriving safety sets from these uncertainty bounds, regardless of whether to include the additional knowledge of the Lipschitz bound (Sui et al., 2015) or not (Berkenkamp et al., 2016), results in potential safety violations. However, since in LoSBO these bounds are ignored for the safe sets, this problem does not occur.

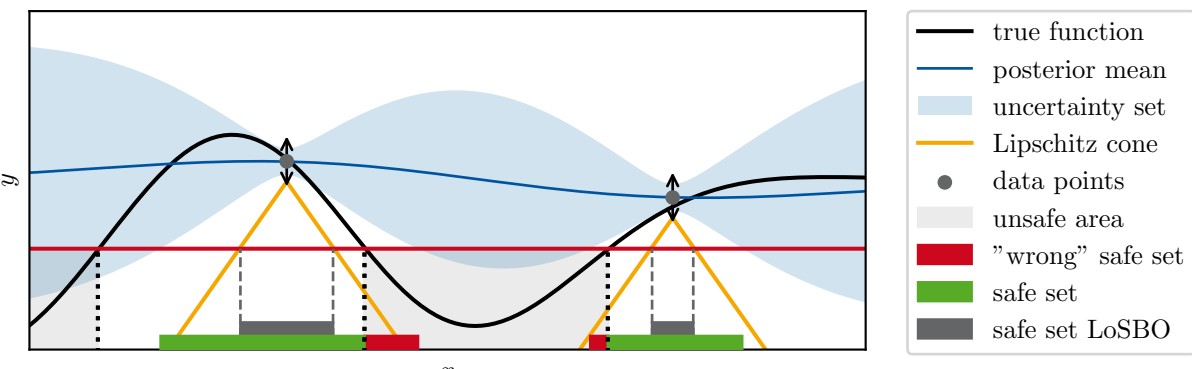

Figure 3: Illustration of LosBO being safe, while a safe set based on invalid uncertainty bounds leads to potential safety violations. The safe set of LoSBO (gray set) is determined by the constant $E$ (black arrow) and the Lipschitz cone (orange). The GP mean and the confidence bounds are illustrated in blue. The points in the safe set given by the lower confidence bound are green if they are safe and red if they are unsafe.

Using a **Lipschitz bound instead of an RKHS norm bound** comes with several advantages. First, a (quantitative) Lipschitz bound has a clear interpretation - it is an upper bound on the slope of the target function. Second, the Lipschitz assumption can be related to established prior knowledge: A known upper bound on the Lipschitz constant corresponds to an a priori bound on the rate of change of a function, i.e., it is related to the sensitivity of the underlying problem. As discussed in-depth in Section 6.1, to the best of our knowledge this is not possible for the RKHS norm in non-trivial cases. Third, as already mentioned above, the Lipschitz assumption is independent of the internal model used for exploration, and hence much less prone to model misspecifications like wrong hyperparameters of kernels. Finally, the exact Lipschitz constant is rarely known in practice, and hence an upper bound has to be used. It is clear that a very conservative upper bound will inhibit exploration, just as a conservative upper bound on the RKHS norm, but this issue appears to be unavoidable with any quantitative a priori assumption.

| Algorithm | SafeOpt | Real-$\beta$-SafeOpt | Real-$\beta$-SafeOpt | Real-$\beta$-SafeOpt | LosBO |
|---|---|---|---|---|---|
| Experimental setting | $\beta = 2$ (heuristic) | $B = 2.5$ (underestimate) | $B = 10$ (correct) | $B = 20$ (overestimate) | $\beta = 2$ |
| Not started % | 1.93 | 3.16 | 30.40 | 68.34 | 0.018 |
| Safety violations % | 3.95 | 0.859 | 0 | 0 | 0 |
| Safety violations worst case % | 28.62 | 13.38 | 0 | 0 | 0 |
| Final performance % | 88.75 | 88.76 | 82.45 | 76.69 | 90.90 |

Table 1: Safety-performance tradeoff in SafeOpt. We evaluated 100 functions sampled from an SE-kernel with $B = 10$. On each function we ran each algorithm 10000 times, starting from two initial safe points.

The original SafeOpt algorithm comes with **conditional**[6] **exploration guarantees**. Since our modification leading to LoSBO essentially separates the safety and exploration mechanisms, the exploration guarantees from SafeOpt are rendered inapplicable in the present context. An inspection of the proof of (Sui et al., 2015, Theorem 1) shows that it cannot easily be modified to apply to LoSBO again, as the argument used there relies on the GP model interacting with the safety mechanism[7]. Furthermore, we suspect that pathological situations exist where LoSBO fails to properly explore. However, we have not observed such a situation in our extensive experiments. While providing (again conditional) exploration guarantees for LoSBO is an interesting focus for future work, we argue that the present lack of such theoretical guarantees does not diminish the relevance and usefulness of this algorithm. First, LoSBO shows excellent exploration performance, as demonstrated in the experiments described in the next section. Second, since the scaling parameters $\beta_t$ (which have an important influence on the exploration performance) are proper tuning parameters in LoSBO, unsatisfying performance of the algorithm can be overcome by using this tuning knob. We would like to stress again that in the previous variants of SafeOpt, this option is not available as the scaling parameters need to lead to valid uncertainty sets.

## 6.4 Experimental evaluation

As discussed in Section 5.2, a meaningful comparison with SafeOpt implementations relying on heuristics is impossible, since the latter essentially address a different problem setting. For this reason, we will compare LoSBO with Real-$\beta$-SafeOpt, which precisely adheres to the original SafeOpt setting. For the empirical evaluations a frequentist approach will be used as this is the most natural setting for SafeOpt-like algorithms, see (Srinivas et al., 2010; Fiedler et al., 2021a;b) for further discussion on this. Unless noted otherwise, in each experiment, we generate 100 RKHS functions (using the pre RKHS and ONB approaches), each with RKHS norm 10, and we compute a (slightly) conservative Lipschitz bound as well as an appropriate initial safe set. Following the previous safe BO literature, we restrict ourselves to compact subsets of $\mathbb{R}^d$, and in this section we use $d = 1$ and $D = [0, 1]$ for simplicity. Furthermore, we use independent additive noise, uniformly sampled from $[-B_\epsilon, B_\epsilon]$, with (5) to compute $\beta_t$ in Real-$\beta$-SafeOpt, and $E = 2B_\epsilon$ and $\beta_t \equiv 2$ in LoSBO. For each function we run the algorithms 10000 times to allow a frequentist evaluation of the behavior. The performance is measured by the (shifted and normalized) gap to the optimal function value $\hat{f}_t^*$. Additional details on the experimental setup are provided in the appendix, cf. Section C.

**Well-specified setting** We start by comparing LoSBO and Real-$\beta$-SafeOpt in a well-specified setting. This means that all the algorithmic parameters are set correctly, in particular, the covariance function used in GP regression is the kernel generating the RKHS from which the target functions are sampled. In Figure 4, the results for this setting are presented. The thick solid lines are the means over all 100 functions and all of their repetitions, the fine lines are the means for each of the individual 100 target functions (over

---

[6]By this we mean that the exploration guarantee given in (Sui et al., 2015, Theorem 1) is conditional on the choice of the kernel. Inspecting the expression for the time $t^*$ in this latter result, we find that this result requires appropriate growth behavior of the maximum information gain $\gamma_t$ in order to lead to non-vacuous exploration guarantees.

[7]More precisely, with LoSBO we cannot ensure that the inequality in the last display in the proof of (Sui et al., 2015, Lemma 7) holds.

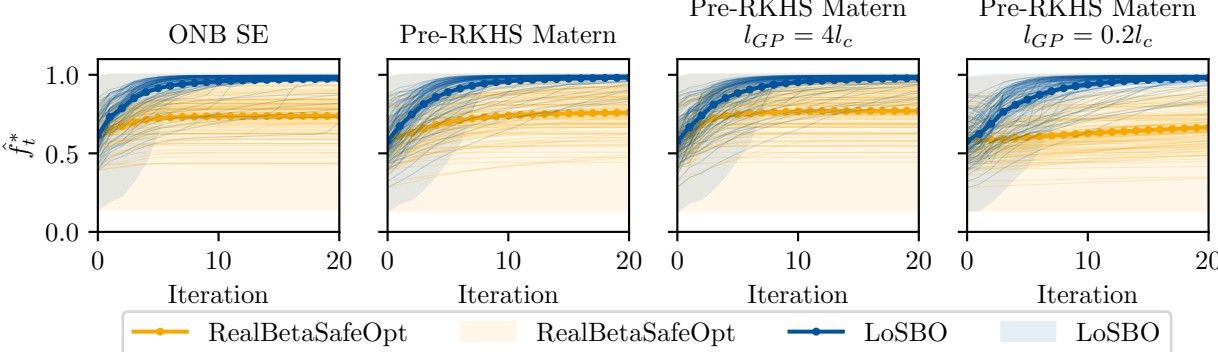

Figure 4: Comparison of LosBO and Real-$\beta$-SafeOpt in a well-specified setting. Thick solid lines are the means over all functions and repetitions, thin solid lines are the means over all repetitions for each individual function, shaded area corresponds to the 10 % and 90 % quantiles over all runs.

10000 repetitions for each function), and the shaded area shows the 10 % and 90 % quantiles, again over all functions. Figure 4, left, displays the results for functions from the Squared Exponential RKHS, sampled using the ONB approach. Interestingly, LoSBO exhibits superior performance compared to Real-$\beta$-SafeOpt, despite providing the latter algorithm with the correct ingredients (Lipschitz bound, kernel, RKHS norm bound, noise variance). Sampling functions using the pre RKHS approach shows only minor differences, cf. Section C.3 in the appendix. Figure 4, second plot, shows the results for RKHS functions corresponding to a Matern-3/2 kernel and sampled with the pre RKHS approach. Qualitatively, we see the same picture, though the performance of both LoSBO and Real-$\beta$-SafeOpt appear to be slightly weaker compared to the previous setting. Intuitively, this is to be expected as Matern RKHS functions are generically less smooth than Squared Exponential RKHS functions, and both LoSBO and Real-$\beta$-SafeOpt rely on a Lipschitz bound, which is in turn related to regularity of functions.

**Misspecified setting** We turn to misspecified settings, where the algorithmic parameters do not match the true setting of the target function. This is particularly interesting in the present situation, since the underlying GP model does not impact the safety of LoSBO, and therefore becomes amenable to tuning. We start with the length scale used in the kernel, which is arguably the most important type of hyperparameter in practice. In Figure 4, third plot, we show the results for overestimating the length scale in GP regression, using Matern kernels. More precisely, a Matern kernel is used both as the kernel for generating the target functions and as a covariance function in GP regression, but the length scale of the covariance function in GP regression is 4 times the ength scale used in the kernel to generate the target function. The qualitative picture remains the same, though it appears that the performance of Real-$\beta$-SafeOpt suffers from the misspecification more than LoSBO. More importantly, in this setting safety violations occur in 12.57 % of all runs. Moreover, for the worst-behaving target function 943 out of 10000 runs lead to safety violations, which is unacceptable in a real-world use case. Figure 4, rightmost plot, shows the complementary situation of underestimating the length scale in GP regression, again using Matern kernels. The length scale of the covariance function used in GP regression is 0.2 times the length scale of the kernel that is used to generate the target function. Again, the qualitative picture remains the same, but the performance degradation is worse for both algorithms in this case. Finally, consider the case where a different kernel is used to generate the target functions than the covariance function in the GP regression. We use a Matern-3/2 kernel to generate the target functions, and a Squared Exponential kernel as covariance function in the GP regression, with the same length scale for both. The results are displayed in Figure 7, lower left. Interestingly, essentially no qualitative difference can be noticed compared to the well-specified Matern case, cf. Figure 7 in the appendix. We suspect that this is due to the correct specification of the length scale, which in the present setting is more important than the kernel misspecification.

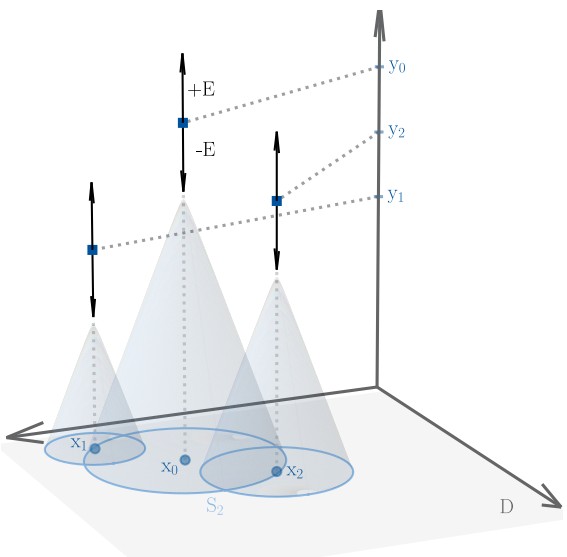 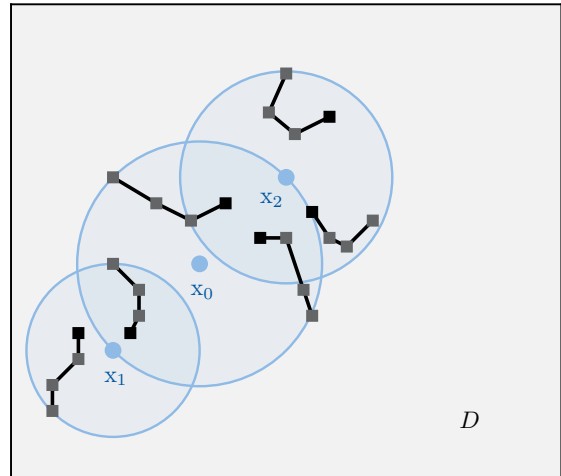

Figure 5: Illustration of safe sets for a 2-dimensional input set. Left: Safe set in LoSBO resulting from three function evaluations. Right: Illustration of the acquisition function optimization in LoS-GP-UCB over a safe set resulting from four function evaluations. Two initial guesses for the local optimization are used for each of the balls that span the safe set according to (12).

## 7   LoS-GP-UCB

In SafeOpt-type algorithms, including our variant LoSBO (Algorithm 3), the central computational step is the optimization of the acquisition function over the expander and maximizer sets, see Algorithm 1. Inspecting the definition of the latter two sets, see Section 3, makes it clear that computing these sets requires a discrete input set $D$. For many typical application scenarios, at least parts of the input set will be continuous (e.g., if the optimization variables include a physical parameter that can vary continuously). This means that some form of discretization is necessary before a SafeOpt-type algorithm can be applicable. Typically, equidistant gridding of the (continuous parts of the) input set is used as a discretization, see (Berkenkamp et al., 2016) for a typical example. As a result, SafeOpt-type algorithms become impractical for even moderate dimensions, e.g., $D \subseteq \mathbb{R}^d$ for $d > 3$ (Kirschner et al., 2019a; Sukhija et al., 2023), and as a member of this class, LoSBO inherits this limitation. In this section, we present and investigate an approach to overcome this issue. Instead of adapting existing solution approaches like (Duivenvoorden et al., 2017; Kirschner et al., 2019a; Sukhija et al., 2023), we suggest a pragmatic and straightforward variant that is motivated by three observations.

*First*, safety in SafeOpt-type algorithms is ensured by restricting the optimization of the acquisition function to (subsets of) safe sets, i.e., sets $S \subseteq D$ such that $f|_S \geq h$, where $f$ is the unknown target function. In other words, as long as we ensure that the acquisition function optimization is restricted to such sets $S$, the resulting algorithm will be safe, no matter how this optimization is performed or whether an additional restriction is added (as in SafeOpt, where the optimization is only over expander and maximizer sets). In the case of LoSBO, these safe sets are of a particularly simple form, as they are the union of closed spheres in a metric space, see Figure 5, left, for an illustration for the case $D \subseteq \mathbb{R}^2$. In typical application scenarios of SafeOpt-type algorithms, the number of input queries is relatively low, and hence the aforementioned union is only over relatively few sets. *Second*, a discrete input set for SafeOpt-type algorithms is necessary due to the involved definition of the expander and maximizer sets, which in turn are defined to guarantee proper exploration in the original SafeOpt setting (Sui et al., 2015). However, for concrete practical problems, such an underexploration might not pose a severe challenge, and an existing BO algorithm can simply be made safe by restricting the optimization of the acquisition function to a safe set $S$ as described above. In fact, the original SafeOpt paper (Sui et al., 2015) already discussed a safe variant of GP-UCB (Srinivas

et al., 2010). This indicates that it might be possible to avoid the complicated sets involved in SafeOpt-type algorithms, and still attain practically useful exploration performance. *Third*, in the current practice of BO, moderately high dimensions of the input space are not problematic for modern BO algorithms[8] Typically, acquisition functions are optimized by running a local optimization method (usually gradient-based) from several initial guesses (which might be random, or based on heuristics). In particular, no gridding is necessary, and this strategy can even deal with moderately high dimensions since local search methods behave well despite increasing dimensionality. In fact, state-of-the-art BO libraries like BOTorch (Balandat et al., 2020) implement exactly this approach as the default option.

Based on these three observations, we now propose a straightforward safe BO algorithm that works even in moderate input dimensions, is compatible with modern BO libraries, and retains all the safety guarantees from LoSBO. In Section 7.1, we describe this algorithm in detail and provide some discussion. In Section 7.2, we evaluate the algorithm empirically and compare it to LoSBO.

### 7.1 Algorithm

Consider the setting of LoSBO as described in Section 6.2. Motivated by the preceding discussion, we start with a standard GP-based BO algorithm that does not need expander and maximizer sets (or similar complicated sets requiring discretization). Due to its relation to SafeOpt, we choose GP-UCB (Srinivas et al., 2010) for this task, so at step $t \geq 1$, the next input is

$$x_{t+1} = \operatorname{argmax}_{x \in D} \mu_t(x) + \beta_t \sigma_t(x), \tag{10}$$

for an appropriate scaling factor $\beta_t \in \mathbb{R}_{>0}$. As usual, ties are broken arbitrarily. Using the (scaled) posterior variance as the acquisition function would be even closer to SafeOpt, but numerical experiments indicate that GP-UCB performs slightly better in this context. Next, we restrict the acquisition function optimization to the safe sets $S_t$ as defined for LoSBO in (9),

$$x_{t+1} = \operatorname{argmax}_{x \in S_t} \mu_t(x) + \beta_t \sigma_t(x). \tag{11}$$

Observe now that

$$S_t = \bigcup_{j=1}^{N_t} \bar{B}_{r_j}(z_j) \tag{12}$$

for some $N_t \in \mathbb{N}_{>0}$, $r_1, \ldots, r_{N_t} \in \mathbb{R}_{>0}$ and $z_1, \ldots, z_{N_t} \in D$, and $\bar{B}_r(z) = \{x \in D \mid d_D(z, x) \leq r\}$ is the closed ball with radius $r \in \mathbb{R}_{>0}$ and center $z \in D$ in the metric space $D$. For example, if the initial safe set has only one element $x_0$ and no input is repeatedly sampled, then $N_t = t + 1$, $z_1 = x_0$ and $z_j = x_{j-1}$ for $j = 2, \ldots, t + 1$. Using the decomposition (12), we now have

$$x_{t+1} = \operatorname{argmax}_{j=1,\ldots,N_t} \operatorname{argmax}_{x \in \bar{B}_{r_j}(z_j)} \mu_t(x) + \beta_t \sigma_t(x). \tag{13}$$

Each of the inner optimization problems $\max_{x \in \bar{B}_{r_j}(z_j)} \mu_t(x) + \beta_t \sigma_t(x)$, $j = 1, \ldots, N_t$, is a maximization problem over the convex sets $\bar{B}_{r_j}(z_j)$, and each of these inner problems are independent. In particular, these optimizations can be trivially parallelized. In practice, one usually has $D \subseteq \mathbb{R}^d$ (often with a simple geometry) and a differentiable covariance function $k$, so it is possible to use a gradient-based local optimization method started from multiple initial guesses. This is illustrated in Figure 5, right. All of these multistarts are independent, and can therefore also be parallelized. We thus arrive at Algorithm 2, which we call Lipschitz-only Safe Gaussian Process Upper Confidence Bound (LoS-GP-UCB) algorithm in the following. LoS-GP-UCB can easily be implemented with state-of-the-art BO libraries. For the numerical experiments described in the next section, we have chosen BoTorch (Balandat et al., 2020), which allows an easy parallel implementation of the acquisition function optimization. Finally, LoS-GP-UCB retains the safety guarantees from LoSBO. In particular, the scaling factors $(\beta_t)_t$ remain tuning factors, and the safety of LoS-GP-UCB is independent of their choice. Furthermore, safety of LoS-GP-UCB *does not require Assumption 2*.

---

[8]Here we are referring to handling the dimensionality within the algorithm, specifically optimization of the acquisition function, and not the exploration performance. Of course, the latter poses challenges, especially if not enough structural or qualitative prior knowledge is encoded in the BO function model.

---

**Algorithm 2** LoS-GP-UCB

---

**Require:** Lipschitz constant $L$, algorithm to compute $\beta_t$, noise bound $E$, initial safe set $S_0$, safety threshold $h$

1: Compute $N_0$, $z_1, \ldots, z_{N_0}$, $r_1, \ldots, r_{N_0}$, $\beta_0$
2: **for** $t = 1, 2, \ldots$ **do**
3:     $x_t = \text{argmax}_{j=1,\ldots,N_{t-1}} \text{argmax}_{x \in \bar{B}_{r_j}(z_j)} \mu_{t-1}(x) + \beta_{t-1}\sigma_{t-1}(x).$         ▷ Determine next input
4:     Query function with $x_t$, receive $y_t = f(x_t) + \epsilon_t$
5:     Update GP with new data point $(x_t, y_t)$, resulting in mean $\mu_t$ and $\sigma_t$
6:     Compute updated $\beta_t$
7:     Compute $N_t$ and add new $z_j$, $r_j$
8: **end for**

---

## 7.2 Experimental evaluation

Analogously to the case of Real-$\beta$-SafeOpt and LoSBO, a frequentist setup will be used, i.e., the algorithm will be run on a fixed target function for many independent noise realizations. In the experiments two aspects will be investigated. First, LoS-GP-UCB will be compared with Real-$\beta$-SafeOpt and LoSBO. Second, we apply LoS-GP-UCB to several benchmark functions with moderate input dimensions.

We start with the comparison to Real-$\beta$-SafeOpt and LoSBO. Since these two algorithms rely on a discrete input space, this comparison necessarily has to be performed on functions with a low dimensional input. We choose essentially the same experimental settings as in Section 7.2 and consider only the well-specified case. As for LoSBO, we use $\beta_t \equiv 2$ in LoS-GP-UCB, and we consider one-dimensional RKHS functions. The algorithms are evaluated on 100 target functions sampled from the pre RKHS corresponding to a Matern-3/2 kernel, and for each function, we run the algorithms 1000 times[9]. The results of this experiment are shown in Figure 6 (a), where we again use the evaluation metric (16). Thick solid lines are the means over all functions and repetitions and shaded areas correspond to the 10 % and 90 % quantiles (again over all functions and realizations). To avoid clutter, the means for each individual function are plotted only for LoS-GP-UCB (thin lines). It is clear from Figure 6 that the performance of Los-GP-UCB is only slightly inferior to the original LosBO algorithm, yet still superior to Real-$\beta$-SafeOpt. This outcome indicates that LoS-GP-UCB is not severely affected by the under-exploration problem described for a safe variant of GP-UCB in (Sui et al., 2015). We suspect that, similar to LoSBO, this is due to the overoptimism resulting from setting $\beta_t \equiv 2$, corresponding to moderately aggressive exploration.

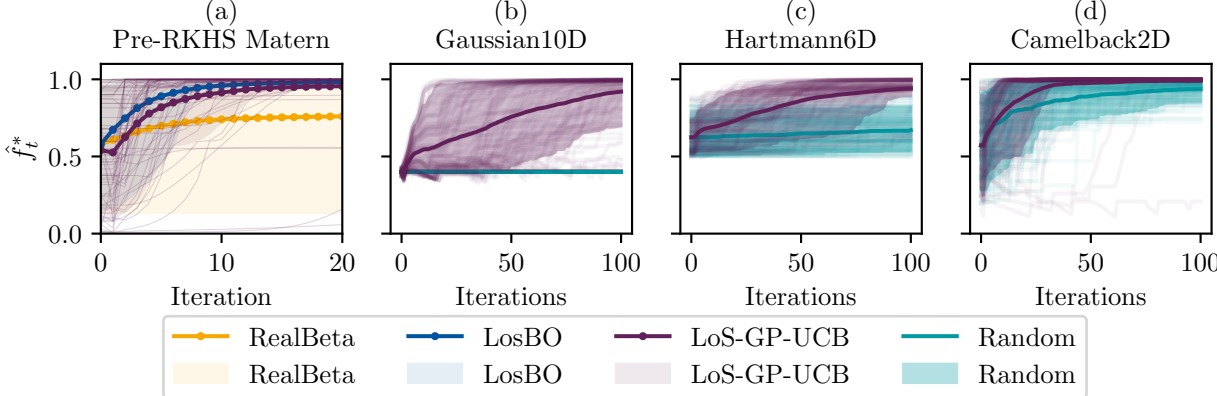

Figure 6: (a) Comparison of LoS-GP-UCB to Real-$\beta$-SafeOpt and LoSBO. (b)-(d) Comparison of Los-GP-UCB to random sampling on more high-dimensional test functions.

---

[9]The qualitative picture is already clear for 1000 repetitions, hence we choose to save computational resources and do not use 10000 repetitions as in Section 7.2.

Let us turn to the evaluation of LoS-GP-UCB on several benchmark functions with moderate to high input dimensions. As test functions, we use the standard benchmarks Camelback (2d) and Hartmann (6d). Similar to (Kirschner et al., 2019a), we also use a Gaussian function $f(x) = \exp\left(-4\|x\|_2^2\right)$ in ten dimensions (10d) as a benchmark. For the Camelback and Hartmann functions, we choose a random initial point in the safe set. For the Gaussian function, we choose a random safe set from the level $f(x_0) = 0.4$. We assume uniformly bounded noise with a noise level of 0.01. In this synthetic setting the Lipschitz constant $L$ is determined by evaluating the function on a fine grid. As a model we use a Squared Exponential kernel with output variance set to 1 and length scale set to $1/L$. For the prior mean we choose 0.5 as the function values are between 0 and 1. Finally, in all these settings, we compare LoS-GP-UCB to random search, and run both algorithms from the same initial safe set for 100 iterations, repeating this 100 times (for different random choices of the initial safe set). The results of this experiment is shown in Figure 6 (b), (c) and (d).

## 8 Conclusion

In this work, we are concerned with practically relevant safety aspects of the important class of SafeOpt-type algorithms. We identified the use of heuristics to derive uncertainty bounds as a potential source of safety violations in the practical application of these algorithms. This prompted us to use recent, rigorous uncertainty bounds in SafeOpt, which allowed us to numerically investigate the safety behavior of this algorithm. We further identified the knowledge of an upper bound on the RKHS norm of the target function as a serious obstacle to reliable real-world applicability of SafeOpt-type algorithms. To overcome this obstacle, we proposed LoSBO, a BO algorithm class relying only on a Lipschitz bound and noise bound to guarantee safety. Numerical experiments demonstrated that this algorithm is not only safe, but also exhibits superior performance. However, analogously to related algorithms, LoSBO is only suitable for low dimensional problems. We therefore also proposed an additional variant (LoS-GP-UCB) suitable for moderately high dimensional problems.

The two key assumptions to ensure safety are a known Lipschitz bound on the target function and a known bound on the additive measurement noise, which have clear interpretations, appear natural in many applications, and are established in domains like control engineering. The safety guarantees of the proposed algorithms rely on these assumptions, and they have to be judged on a case-by-case base by practitioners.

Ongoing work is concerned with implementing the presented algorithms for safe learning in an automotive context. While this article focuses on safety, providing exploration guarantees for LoSBO is an interesting aspect of future work. We expect that the approach outlined in Section 6 applies to most SafeOpt variants. The derivation, implementation, and evaluation of the corresponding LoSBO-type algorithms for these variants is thus another interesting direction for future work. Our findings in combination with evidence in the literature that SafeOpt and related algorithms have been successfully used in various applications indicate that this algorithm class does not ensure hard safety constraints (in practice), but instead yields "cautious" behavior. The precise connection to conservative bandits and existing cautious BO approaches is another interesting topic for further investigations.

### Broader Impact Statement

This work is concerned with safety issues of a popular BO algorithm that has already found numerous applications in real-world scenarios. Henceforth we contribute to the improved safety and reliability of machine learning methods for real-world applications. Furthermore, we expect no adverse societal impact of our work.

### Acknowledgments

We thank Paul Brunzema and Alexander von Rohr for very helpful discussions and Sami Azirar for support when generating plots. This work was performed in part within the Helmholtz School for Data Science in Life, Earth and Energy (HDS-LEE). Furthermore, the research was in part funded by the German Federal Ministry for Economic Affairs and Climate Action (BMWK) through the project EEMotion and by the Deutsche Forschungsgemeinschaft (DFG, German Research Foundation) under Germany's Excellence Strategy EXC-

2023 Internet of Production 390621612. Computations were performed with computing resources granted by RWTH Aachen University under project rwth1459.

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

# A Background on RKHSs

## A.1 A brief introduction

Consider a function $k : D \times D \to \mathbb{R}$. We call $k$ *positive semidefinite* if for all $x_1, \ldots, x_N \in D$, $N \in \mathbb{N}_{>0}$, the matrix $\big(k(x_i, x_j)\big)_{i,j=1,\ldots,N}$ is positive semidefinite. Equivalently, the function $k$ is symmetric ($k(x_i, x_j) = k(x_j, x_i)$ for all $i, j = 1, \ldots, N$), and for all $\alpha_1, \ldots, \alpha_N \in \mathbb{R}$ we have $\sum_{i,j=1}^{N} \alpha_i \alpha_j k(x_i, x_j) \geq 0$. In the literature such a function is often called *positive definite* or *of positive type*. Additionally, $k$ is called *positive definite*, if for all pairwise distinct $x_1, \ldots, x_N \in D$, the matrix $\big(k(x_i, x_j)\big)_{i,j=1,\ldots,N}$ is positive definite. This property is sometimes referred to as *strict positive definiteness* in the literature.

Let $H$ be a Hilbert space functions on $D$. We call $H$ a *reproducing kernel Hilbert space (RKHS)* if every evaluation functional is continuous, i.e., for all $x \in D$ the mapping $H \ni f \mapsto f(x) \in \mathbb{R}$ is continuous w.r.t. the norm induced by the scalar product in $H$. Furthermore, $k$ is called a *reproducing kernel (for $H$)* if 1) $k(\cdot, x) \in H$ for all $x \in D$, and 2) $f(x) = \langle f, k(\cdot, x) \rangle_H$ for all $f \in H$ and $x \in D$.

As is well-known, $H$ is a RKHS if and only if it has a reproducing kernel, and in this case the latter is unique (Steinwart & Christmann, 2008, Lemma 4.19, Theorem 4.20). Furthermore, every reproducing kernel is positive semidefinite, and every positive semidefinite function is a reproducing kernel for a unique RKHS (Steinwart & Christmann, 2008, Theorem 4.16, 4.21). If $k$ is positive semidefinite, then we denote its unique RKHS as $(H_k, \langle \cdot, \cdot \rangle_k)$, and the induced norm by $\| \cdot \|_k$. Furthermore, we define the *pre RKHS* by

$$H_k^{\mathrm{pre}} = \mathrm{span}\,\{k(\cdot, x) \mid x \in D\} = \left\{ \sum_{n=1}^{N} \alpha_n k(\cdot, x_n) \mid N \in \mathbb{N}_{>0}, \alpha_n \in \mathbb{R}, x_n \in D,\, n = 1, \ldots, N \right\}, \quad (14)$$

and this subspace is dense in $H_k$ w.r.t. $\| \cdot \|_k$. Given $f = \sum_{n=1}^{N} \alpha_n k(\cdot, x_n)$, $g = \sum_{m=1}^{M} \beta_m k(\cdot, y_m) \in H_k^{\mathrm{pre}}$, we have

$$\langle f, g \rangle_k = \sum_{n=1}^{N} \sum_{m=1}^{M} \alpha_n \beta_m k(y_m, x_n), \quad (15)$$

see (Steinwart & Christmann, 2008, Theorem 4.21).

If $k$ is the covariance function of a GP, then it is positive semidefinite (since every covariance matrix is positive semidefinite), and hence the reproducing kernel of a unique RKHS. Conversely, if $k$ is the reproducing kernel of a RKHS, then $k$ is positive semidefinite, there exists a GP having $k$ as its covariance function, and the GP can be chosen with a zero mean function (Berlinet & Thomas-Agnan, 2004).

Furthermore, consider GP regression with a prior $\mathcal{GP}_D(m, k)$, then $\mu_t - m \in H_k^{\mathrm{pre}}$, where $\mu_t$ is the posterior mean corresponding to a finite data set with $t$ points. In particular, the posterior mean for a zero mean prior GP is always in the pre RKHS corresponding to the covariance function. As is customary in machine learning with GPs (Rasmussen & Williams, 2006), and also in many BO scenarios (Garnett, 2023), in the following we will use a zero mean GP prior, $m \equiv 0$, without loss of generality.

## A.2 Additional discussion

**Theory for the RKHS norm** For an arbitrary kernel, one can use discretization-based variational characterizations of the RKHS norm (and RKHS functions), for example, by maximization over a family of lower bounds on the RKHS norm (Fiedler et al., 2024, Section B), (Atteia, 1992, Chapter I), by minimization over certain bounds on function values at finitely many inputs (Okutmuştur, 2005, Theorem A.2.6), by minimization over finite interpolation problems (Paulsen & Raghupathi, 2016, Theorem 3.11), or by minimization over certain matrix inequalities (Paulsen & Raghupathi, 2016, Theorem 3.11). For separable RKHSs, the RKHS norm can be expressed using a sampling expansion (Korezlioglu, 1968), or as the limit of norms of RKHSs over finite inputs (Lukić & Beder, 2001, Lemma 4.6). On the one hand, all of these variational problems have an explicit form and they work for *any* kernel (any kernel with separable RKHS, respectively). However, it is not at all clear how to relate these representations to common properties of functions that might be used as reliable prior knowledge to derive upper bounds on the RKHS norm. Furthermore, these variational

problems generally cannot be used in numerical methods to estimate upper bounds on the RKHS norm, but only lower bounds, though they may be used for estimating bounds in heuristics (Tokmak et al., 2023). Since these characterizations are based on discretizations of a given RKHS function, in particular, using the exact function values, they are not suitable in the present setting of unknown target functions only accessible through noisy evaluations.

If one considers more specific classes of kernels, other characterizations of the RKHS norm become available. For example, continuous kernels on a compact metric space equipped with a measure having full support (often called a Mercer kernel in this context) allow a description of the RKHS norm as a weighted $\ell_2$-norm (Steinwart & Christmann, 2008, Section 4.5), based on Mercer's theorem. This has a clear interpretation in the context of kernel methods, in particular, giving insight into the regularization behavior of the RKHS norm in optimization problems in kernel machines (Hastie et al., 2009, Section 5.8), which in turn can be used to derive learning rates for various statistical learning problems (Steinwart et al., 2009). More general forms of Mercer's theorem are available (Steinwart & Scovel, 2012), which in turn lead to improved learning theory results (Fischer & Steinwart, 2020). While the RKHS norm representation for Mercer kernels is an important tool for statistical learning theory and provides intuition about the regularization behavior, it is again unclear how it can be used to derive *quantitative* RKHS norm bounds. Expressing the RKHS norm for Mercer kernels as a weighted $\ell_2$-norm provides valuable *qualitative* intuition about the corresponding RKHS norm, but we are not aware of any practically relevant example where this has been used to translate realistic prior knowledge into a concrete upper bound on the RKHS norm.

Similarly, for sufficiently regular translation-invariant kernels, the RKHS norm can be expressed as a weighted integral over the Fourier transform of RKHS functions (Wendland, 2004, Theorem 10.12). This formulation allows an intuitive interpretation of the RKHS norm as a generalized energy, penalizing high-frequency behavior of RKHS functions (as determined by the Fourier transforms of the kernel). Several important function spaces are related to RKHSs, for example certain Sobolev spaces (Wendland, 2004, Chapter 10) or Fock spaces (Steinwart & Christmann, 2008, Section 4.4), which again have their own representations of the RKHS norm (potentially after some embedding). Again, all of these representations offer insights into the RKHS norm, and are important theoretical tools, but how this can be used to derive practically useful quantitative upper bounds on the RKHS norm remains unclear.

To summarize, while an extensive body of work on characterization and representation results for the RKHS norm is available, these results appear to be unsuitable to derive numerical upper bounds on this norm using practically meaningful prior knowledge.

**RKHS membership** The assumption that the target function $f$ is contained the RKHS $H_k$ is standard in the BO literature (Abbasi-Yadkori, 2013; Srinivas et al., 2010; Chowdhury & Gopalan, 2017; Whitehouse et al., 2024), where $k$ is a kernel that is used as the covariance function in GP regression. In general, if functions from $H_k$ are to be used as models of the underlying target function, the RKHS $H_k$ has to be sufficiently rich. In many cases, as a qualitative assumption, this is quite mild. For example, if $D$ is a compact metric space, then a large variety of *universal kernels* are available – kernels with an RKHS that is dense (w.r.t. the supremum norm) in the set of all continuous functions on $D$ (Steinwart & Christmann, 2008, Section 4.6). However, the assumption $f \in H_k$ is considerably more stringent and delicate. Specifically, even containment of the target function within an RKHS generated by a kernel from the same class as the one used as a covariance (e.g., a squared exponential or Matern kernel) may not be sufficient, because there can be a mismatch of hyperparameters. For the case of squared exponential kernels, a complete characterization of the inclusions of RKHSs w.r.t. different hyperparameters is described in (Steinwart & Christmann, 2008, Section 4.4), revealing that this situation can indeed occur. While there is work on covariance function misspecification in GP regression (Wynne et al., 2021)[10] and GP-based BO (Berkenkamp et al., 2019; Bogunovic & Krause, 2021), these works do not help in the present situation, because of the form of the resulting guarantees in the former case, and the potential for schemes sampling unsafe inputs in the latter case. The situation is further complicated by the fact that in practice the hyperparameters are adapted during BO (Garnett, 2023), theoretical guarantees of which are only slowly emerging (Teckentrup, 2020; Karvonen et al., 2020; Karvonen & Oates, 2023). While these issues do not directly apply to LoSBO

---

[10]Even likelihood misspecification.

and LoS-GP-UCB, cf. also the discussion in Section 6.3, they are relevant for Real-$\beta$-SafeOpt. In Sections 5 and 6, we work (unless noted otherwise) with the assumption $f \in H_k$, and do not consider hyperparameter adaption during the optimization process. This is justified from two perspectives. First, it is a common approach in the BO literature, including safe BO (Sui et al., 2015; 2018). Second, and more importantly, the safety-related issues that we are concerned with in this work are independent of this issue (though of course, it is an additional problematic factor).

## B    Additional implementation details and practical considerations

**Pseudocode for LoSBO**    For the reader's convenience, we provide a detailed description of LoSBO in Algorithm 3. As discussed in Section 6, the only change compared to SafeOpt (and Real-$\beta$-SafeOpt) is in the computation of the safe sets.

---

**Algorithm 3** LoSBO

**Require:** Lipschitz constant $L$, algorithm to compute $\beta_t$, noise bound $E$, initial safe set $S_0$, safety threshold $h$

1: $Q_0(x) \leftarrow \mathbb{R}$ for all $x \in D$                                            ▷ Initialization of uncertainty sets
2: $C_0(x) \leftarrow [h, \infty)$ for all $x \in S_0$
3: $C_0(x) \leftarrow \mathbb{R}$ for all $x \in D \setminus S_0$
4: **for** $t = 1, 2, \ldots$ **do**
5:     $C_t(x) \leftarrow C_{t-1}(x) \cap Q_{t-1}(x)$ for all $x \in D$   ▷ Compute upper and lower bounds for current iteration
6:     $\ell_t(x) \leftarrow \min C_t(x)$, $u_t(x) \leftarrow \max C_t(x)$ for all $x \in D$
7:     **if** $t > 1$ **then**                                                                      ▷ Compute new safe set
8:         $S_t = S_{t-1} \cup \{x \in D \mid y_{t-1} - E - Ld(x_{t-1}, x) \geq h\}$
9:     **else**
10:         $S_1 = S_0$
11:     **end if**
12:     $G_t \leftarrow \{x \in S_t \mid \exists x' \in D \setminus S_t : u_t(x) - Ld(x, x') \geq h\}$        ▷ Compute set of potential expanders
13:     $M_t = \{x \in S_t \mid u_t(x) \geq \max_{x_S \in S_t} \ell_t(x_S)\}$        ▷ Compute set of potential maximizers
14:     $x_t \leftarrow \arg\max_{x \in G_t \cup M_t} w_t(x)$                                          ▷ Determine next input
15:     Query function with $x_t$, receive $y_t = f(x_t) + \epsilon_t$
16:     Update GP with new data point $(x_t, y_t)$, resulting in mean $\mu_t$ and $\sigma_t$
17:     Compute updated $\beta_t$
18:     $Q_t(x) = [\mu_t(x) - \beta_t \sigma_t(x), \mu_t(x) + \beta_t \sigma_t(x)]$ for all $x \in D$
19: **end for**

---

**Choice of noise bound in LoSBO**    Proposition 1 states that if $E \in \mathbb{R}_{>0}$ is a bound on the noise magnitude, then LoSBO is safe. Suppose we know that $|\epsilon_t| \leq B_\epsilon$ for some constant $B_\epsilon \in \mathbb{R}_{>0}$, then we could set $E = B_\epsilon$, and assume that the bound $B_\epsilon$ is sharp. For example, we might have $\epsilon_t \sim \frac{1}{2}\delta_{B_\epsilon} + \frac{1}{2}\delta_{-B_\epsilon}$, where $\delta_x$ is the Dirac distribution with atom on $x$. If we choose $E = B_\epsilon$, then LoSBO is indeed safe according to Proposition 1, i.e., for all inputs $x_t \in D$ that are queried by the algorithm, we have $f(x_t) \geq h$. However, if we additionally assume that there are sizeable parts of the input space $D$ where $f \approx h$, and since the border of the safe sets will be in such a region, it is likely that inputs from this area will be queried. This means that it is likely that measurements $y_t$ with $y_t < h$ will be received - this happens if $f(x_t) \approx h$ and $\epsilon_t \approx -B_\epsilon$. While according to the formal model such an input $x_t$ is safe, to the user it looks as if a safety violation occurred. Since in practice a detected (not necessarily real) safety violation might lead to some costly action (e.g., emergency stop), such a situation is undesirable. To avoid this, we can set $E = 2B_\epsilon$. While this introduces some conservatism, it avoids the described apparent safety violations - essentially, this option mitigates false alarms. Whether to choose in the present situation $E = B_\epsilon$ or $E = 2B_\epsilon$ (or even an option in between) is ultimately a practical question to be addressed by the practitioner using the algorithm.

**Assumptions of LoSBO**    Inspecting the proof of Proposition 1 reveals that considerably more general (and weaker) assumptions can be used with the same argument.

1. Instead of a fixed noise bound $E$, one can mutatis mutandis use asymmetric, time-varying and heteroscedastic noise. Formally, one can assume that two functions $E_\ell, E_u : D \times \mathbb{N}_0 \to \mathbb{R}_{\geq 0}$ exist, such that for all $t \geq 1$ and $x \in D$, it holds that $E_\ell(x, t) \leq \epsilon_t \leq E_u(x, t)$, if $x$ is the input used at time $t$.

2. Instead of Lipschitz continuity, one can assume that there exists a continuous and strictly increasing function $\phi : \mathbb{R}_{\geq 0} \to \mathbb{R}_{\geq 0}$ with $\phi(0) = 0$, such that for all $x, x' \in D$ it holds that $f(x') \geq f(x) - \phi(d_D(x, x'))$. This includes the case of Hölder continuity, which has previously been used in a similar context (Calliess, 2014).

To keep the presentation focused and easy to follow, we do not use this additional flexibility in the present work, but all developments in Sections 6 and 7 applies to these more general cases.

## C    Additional material on experiments

### C.1    Additional details for Section 5.1

In Section 5.1, we work exclusively with the squared exponential (SE) kernel on the input set $[0, 1]$ with length scale $0.2/\sqrt{2}$. In order to generate a synthetic RKHS function with known RKHS norm, we follow the approach from Fiedler et al. (2021a) and use the explicit orthonormal basis (ONB) of the SE kernel RKHS as described in (Steinwart & Christmann, 2008, Section 4.4). More precisely, we select a random subset of these basis functions, generate randomly a vector of coefficients and rescale it (to achieve the required RKHS norm), and then form the weighted sum of the selected basis functions.

For the first experiment, in order to generate the data sets, 100 inputs are uniformly sampled from $[0, 1]$, the corresponding RKHS function is evaluated on these inputs, and finally i.i.d. normal noise with variance 0.01 is added to the function values. For each data set, we apply GP regression with a zero mean prior, and the SE kernel as covariance function, using the same length scale as for the generation of the target functions, and we use the actual noise variance in the GP regression. Finally, we use the uncertainty set (2) with $\beta_t \equiv 2$, and check on a fine grid on $[0, 1]$ whether the target function from the RKHS is completely contained within this uncertainty set. As soon as the function value of one input is not covered by the uncertainty, we count this run as a failure. Note that this might be even slightly optimistic, since a bound violation might happen in the between the grid points (which is very unlikely in this setup, however).

For the second experiment, we use the same approach as above to generate RKHS functions $f$. We define the safety threshold $h$ by setting $h = \hat{\mu}(f) - 0.2\widehat{SD}(f)$, where $\hat{\mu}(f)$ and $\widehat{SD}(f)$ are the empirical mean and standard deviation of the test function $f$ evaluated on a fine grid of the input space. This procedure is necessary to ensure that we get non-empty safe sets, but that the problem instance is sufficiently challenging. We further evaluate $|f'|$ on a fine grid on the input space, take the maximum of this, and multiply it by 1.1 to find a (slightly conservative) upper bound of the Lipschitz constant of $f$, which is used as the Lipschitz bound in the algorithm. We then run SafeOpt on 10000 times from a random safe initial state, again using i.i.d. additive normal noise with variance 0.01 (and using the correct noise variance in the GP regression). Finally, we select a target function that exhibits significant safety violations. Note that this approach is appropriate in this context, since the safety property has to hold for *all* potential target functions, and due to the large number of independent runs, no multiple testing problem arises.

### C.2    Additional details for Section 6.4

A target function is therefore fixed, and the algorithms are run multiple times on this same function with independent noise realizations. To enable the performance of the algorithms to be clearly evaluated, synthetic target functions will be used, and since we want to compare LoSBO with Real-$\beta$-SafeOpt - the latter requiring a target function from an RKHS and with a known RKHS norm upper bound - we generate target functions from an RKHS. Unless noted otherwise, in each experiment we generate 100 RKHS functions, each with RKHS norm 10, and we compute a (slightly) conservative Lipschitz bound as well as an appropriate initial safe set. Following the previous safe BO literature, we restrict ourselves to compact subsets of $\mathbb{R}^d$, and in this section for simplicity we further restrict ourselves to $d = 1$.

The frequentist setup is inherently worst-case, but for numerical experiments it is necessary to restrict ourselves to finitely many RKHS functions. Nevertheless, the RKHS functions used should be somewhat representative to give a meaningful indication of the algorithmic performance. Specifically, any bias due to the function generating method should be minimized. In the following experiments, we sample functions from the pre RKHS, i.e., given a kernel $k$, we randomly choose some $M \in \mathbb{N}_{>0}$, $\alpha_1, \ldots, \alpha_M \in \mathbb{R}$ and $x_1, \ldots, x_M \in D$ and then use $f = \sum_{i=1}^{M} \alpha_i k(\cdot, x_i) \in H_k$ as a target function, which works for *any* kernel. In the case of the squared exponential kernel, we also utilize the ONB described in Steinwart & Christmann (2008, Section 4.4), which we have already used for some of the experiments in Sections 5.1 and 6.1. Generating RKHS functions with more than one method ensures more variety of the considered RKHS functions. Moreover, with both approaches, the exact RKHS norm is available (and can be set by normalization), and the generated functions can be evaluated at arbitrary inputs. Unless noted otherwise, we generate RKHS functions with an RKHS norm of 10, i.e., we consider target functions $f \in H_k$ with $\|f\|_k = 10$. For a more thorough discussion of generating RKHS functions and subtle biases due to the chosen method, we refer to (Fiedler et al., 2021a).

LoSBO and Real-$\beta$-SafeOpt work on arbitrary metric spaces, as long as a kernel can be defined on that space. Following the previous safe BO literature, we restrict ourselves to compact subsets of $\mathbb{R}^d$, and in this section for simplicity we further restrict ourselves to $d = 1$. To run LoSBO and Real-$\beta$-SafeOpt, we need a bound on the Lipschitz constant of the target function, as well as an initial safe set. For the former, we restrict ourselves to kernels inducing continuously differentiable RKHS functions, since the latter are Lipschitz continuous due to the compact domain. To determine a bound on the Lipschitz constant, we evaluate the target function on a fine discretization of the input domain, numerically compute an approximation of the Lipschitz constant, and multiply the result by 1.1 to counterbalance the discretization error. Since the target functions are randomly generated, we compute an appropriate safety threshold for each function so that some portions of the input space are safe, and some are unsafe. This avoids trivial situations for safe BO. More precisely, for a given target function $f$, we compute its empirical mean $\hat{\mu}(f)$ and empirical standard deviation $\widehat{SD}(f)$ on a fine grid, and then set $h = \hat{\mu}(f) - 0.2\widehat{SD}(f)$. Next, for each target function $f$ and safety threshold $h$, we need to generate an initial safe set. Similar to the choice of the safety threshold, trivial situations should be avoided, particularly cases where no safe exploration is possible. To achieve this goal, we first determine some $x_0 \in \arg\max_{x \in D} f(x)$, then consider the set $D \cap I_{x_0}$, where $I_{x_0}$ is the largest interval such that $x_0 \in I_{x_0}$ and $f|_{I_{x_0}} \geq h + E$. Finally, one input is randomly selected from this set, and the singleton set containing this latter input is then selected as the initial safe set. Using a singleton initial safe set is common in the literature on SafeOpt-type algorithms, see (Berkenkamp et al., 2016).

The typical application scenario for SafeOpt-type algorithms is the optimization of some performance measure by interacting with a physical system. In particular, each function query is relatively expensive, hence in these scenarios only a few function values are sampled. Motivated by this, in all of the following experiments, for each target function, LoSBO and Real-$\beta$-SafeOpt are run for 20 iterations, starting from the same safe set. For each target function, this is repeated 10000 times to allow a frequentist evaluation of the behavior. Finally, each type of experiment is run with 100 different randomly generated target functions. To make runs with different target functions comparable, we evaluate the performance in a given run of a target function $f$ by

$$\hat{f}_t^* = \frac{f\left(\arg\max_{x \in S_t} \mu_t(x)\right) - h}{f^* - h}, \tag{16}$$

where $\mu_t(x)$ is the predictive mean (in LoSBO and Real-$\beta$-SafeOpt the posterior mean) at time $t \geq 1$, evaluated at input $x$, and $f^*$ is the maximum of $f$. This metric will be averaged over all runs for a given $f$, and over all target functions, respectively, in the following plots.

For simplicity, independent additive noise, uniformly sampled from $[-B_\epsilon, B_\epsilon]$, is used in all of the following experiments. As is well-known, bounded random variables are subgaussian, and we can set $R = B_\epsilon$ in Real-$\beta$-SafeOpt. Additionally, we choose $\delta = 0.01$ and the true RKHS norm as the RKHS norm upper bound in Real-$\beta$-SafeOpt, unless noted otherwise. We further set the nominal noise variance equal to $R$ in both LoSBO and Real-$\beta$-SafeOpt. Following the discussion in Section B, we choose $E = 2B_\epsilon$ in LoSBO. Finally, we must specify a strategy to compute $\beta_t$ in LoSBO. Recall from Section 6.2 that these scaling factors are now proper tuning parameters. In all of the following experiments, we use $\beta \equiv 2$ in LoSBO, as this is a common choice in the literature on SafeOpt and GP-UCB. Choosing such a simple rule also simplifies the

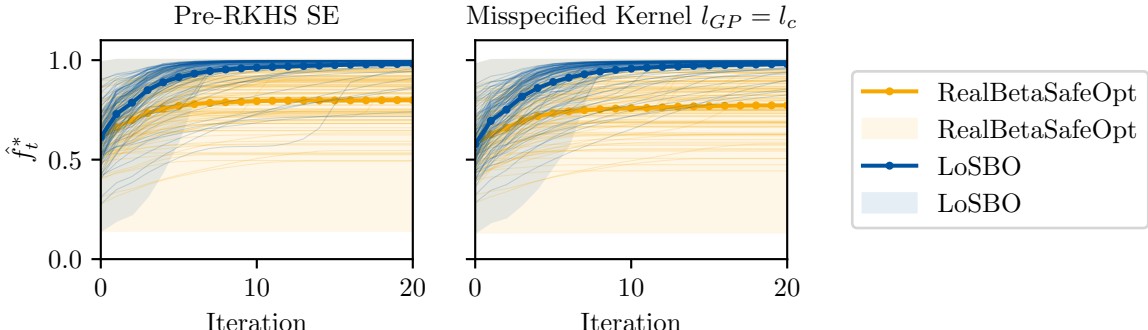

Figure 7: Additional experimental results for LoSBO. Left: Well-specified setting using the SE covariance function and sampling target functions using the pre RKHS approach. Left: Misspecified covariance function.

experimental evaluation, as no additional tuning parameters or further algorithmic choices are introduced. Unless noted otherwise, in all of the following experiments $B_\epsilon = 0.01$ is used.

### C.3 Additional experimental results for LoSBO

We have repeated the first experiment from Section 6.4, however, now with the RKHS functions sampled using the pre RKHS approach. Figure 7, left, displays the results. While no real difference in performance is noticeable for LoSBO, Real-$\beta$-SafeOpt appears to perform slightly better compared to target functions generated using the ONB approach. A potential explanation lies in the shapes of functions that typically arise in the two different sampling methods. As observed in (Fiedler et al., 2021a), functions sampled from the ONB look more "bumpy" compared to pre RKHS functions, and appear to be more challenging for the uncertainty bounds. Since Real-$\beta$-SafeOpt needs to precisely adhere to these bounds, its exploration performance is diminished. By contrast, LoSBO behaves overly optimistically as $\beta_t \equiv 2$ is used, but the underlying RKHS functions have RKHS norm 10, see also the evaluations in (Fiedler et al., 2021a). It appears that this over-optimism leads to better performance, and since for safety LoSBO *does not* rely on scaling factors $\beta_t$ that correspond to valid frequentist uncertainty sets, this over-optimism does not jeopardize safety.

