# OpenReview forum: "On Safety in Safe Bayesian Optimization"
_TMLR — Accepted by TMLR_

### Review · Reviewer_Eqfg · 2024-07-05

**Summary Of Contributions:**

The authors identify and analyze three problems with current Safe Bayesian Optimization (SafeOpt) algorithms, propose three novel SafeOpt algorithms - one to address/solve each problem. First, the authors demonstrate that using typical Safe BO algorithms often leads to safety guarantees being invalid in practice. To solve this problem, the authors propose a new algorithm (the Real-β-SafeOpt algorithm) which only requires relatively small changes to the standard SafeOpt algorithms to recover the desired theoretical safety guarantees. Secondly, the authors point out that many SafeOpt algorithms are difficult to apply in real-world settings since they make the assumption that the target function has an upper bound on the reproducing kernel Hilbert space (RKHS) norm. To solve this problem, the authors introduce an additional new SafeBO algorithm (the Lipschitz-only Safe BO algorithm) which maintains safety guarantees without the need for the limiting RKHS norm assumption. In addition to removing this assumption, the authors demonstrate that this algorithm outperforms other state-of-the-art SafeBO algorithms on several test functions. Third, the authors point out that SafeOpt and other similar algorithms are difficult to apply to higher-dimensional problems since they assume a discrete search space. To fix this, the authors propose a third algorithm (Lipschitz-only Safe GP-UCB) which is makes Safe BO applicable to higher-dimensional functions while still having the same safety guarantees.

**Audience:**

Yes

**Claims And Evidence:**

Yes

**Requested Changes:**

None of the changes list below are critical to securing my recommendation for acceptance of the paper, but I do think they could strengthen the work.
1.	The authors claim that the significance of LoS-GP-UCB is that it allows SafeBO algorithm to be applied to higher-dimensional functions. However, in the experimental results section, the highest dimensional function that the authors apply LoS-GP-UCB to is 10-dimensional. It is still exciting and significant that LoS-GP-UCB is able to scale SafeBO up to 10 dimensions! But it would be very interesting to see what the limit is on the dimensionality where LoS-GP-UCB can be applied successfully? Could it be applied to say 20-dimensional function? How about 100-dimensional functions? Additional experiments on higher-dimensional functions could show what the upper limit is on the dimensionality where LoS-GP-UCB is no longer very useful. I think this would be informative for readers who may want to use this algorithm in practice, but again, this addition is not critical to securing my recommendation for acceptance.
2.	 While the text is clear and easy to follow, some of the text could likely be made more concise (i.e. some things are said in several sentences that could be condensed down to one sentence).

**Strengths And Weaknesses:**

Strengths:
1.	The paper is well motivated since it is clear to me that each problem the authors identify with current SafeOpt algorithms is important to solve.
2.	Very good use of diagrams to illustrate points/algorithms.
3.	Algorithm blocks make it very clear which each algorithm is doing and imporves readability.
4.	The paper is well written and easy to follow.

Weaknesses:
1.	See in requested changes below

---

> ### Author Response · Authors · 2024-07-27
>
> Thank you very much for your detailed review and positive evaluation of our work.
>
> **Requested changes**
> Item 1
> This is an excellent point. LoS-GP-UCB is based on GP-UCB and an implementation strategy used in practice (notably, in BoTorch), so it inherits the limitations of these. In particular, this approach is in general only suitable for moderate dimensional problems (as considered in our experiments). It is indeed a very interesting question to investigate when it fails (e.g., by running it on batches of test functions of increasing input dimensions), and how to deal with truly high-dimensional problems (from the perspective of BO, so say 100 dimensional). However, in our opinion these (albeit very interesting) questions are beyond of the scope of the present work. In particular, dealing with high dimensional (in contrast to "just" moderate dimensional) problems requires new considerations (e.g., how to include certain structural assumptions on the underlying target function, or moving to a local search setting) and techniques (e.g., using random projection-type methods) which are orthogonal to the present work.
>
> Item 2
> >While the text is clear and easy to follow, some of the text could likely be made more concise (i.e. some things are said in several sentences that could be condensed down to one sentence).
>
> This is an excellent suggestion, we will work on the manuscript to make it more concise without reducing clarity and precision.

---

### Review · Reviewer_QF7s · 2024-07-07

**Summary Of Contributions:**

This paper studies "safe" Bayesian optimization (BO), which I understand is maximizing a function $f(x)$ while avoiding the evaluation of any inputs $x$ where $f(x)<h$ for some constant $h\in\mathbb{R}$.
The paper describes some theoretical and practical limitations of existing algorithms for safe BO,
then proposes a new algorithm called LoSBO (Lipschitz-only safe BO) which uses a Lipschitz continuity assumption to guarantee the "safety" behavior.

**Audience:**

Yes

**Claims And Evidence:**

Yes

**Requested Changes:**

The main change I would request is to add discussion contrasting the pros/cons of estimating RKHS norm bounds vs Lipschitz constant bounds. Also, acknowledging the failure mode of underexploring if the Lipschitz constant is high is also important to guide future users of the proposed algorithms.

I also highly highly suggest re-writing the paper to be much shorter (ideally < 10 pages). To do this, you can look at my notes above and think about re-structuring the paper by merging semi-duplicated sections and moving things to appendices. Here are some additional notes:
- No need to repeatedly say how important safety is: you can just mention it once in the intro.
- You can use appendices to add content which could be helpful to the paper, but not important for the main narrative. Here are some things which could go into an appendix?
  - Experiments where existing BO algorithms are not safe if the parameters are not tuned correctly (it should not be surprising that this can happen; you can just mention your constructed example and refer readers to the appendix if they are interested in the details.)
  - Extended definitions of RKHSs
  - Extended literature review (where not super related to main point of paper)
- Keep remarks short- e.g. remark 1 is not really a "remark" in my opinion... ([reading](https://math.stackexchange.com/questions/2097999/in-writing-mathematical-papers-what-qualifies-as-a-remark-and-not-just-some-d?rq=1))
- Clarify what input dimension was used for your experiments in sections 6-7 (this detail was not clear)

Also, small typos:
- You use ">>" instead of "\gg" to type $\gg$
- The notation "SD" with a hat looks awkward. You probably don't know the `\widehat{}` command.
- in your citations, you cite "hernandez et al" when it should be "Hernández-Lobato et al" (note the accents and a hyphenated last name)

**Strengths And Weaknesses:**

Overall this paper has some good critiques of previous safe BO methods and proposes a reasonable algorithm. That being said, after a lot of criticism of previous papers for unrealistic assumptions of bounding constants/etc, the authors provide a pretty flimsy justification for why we would know the Lipschitz constant of the function being optimized. Furthermore, for what it provides, the paper is too long in my opinion- it feels like a good 6-page paper trapped in the body of a 24 page paper. I think the authors can cut a lot out. More on these points below.

## Strength 1: criticisms

I like the analysis of the following criticisms:
- Previous works not setting $\beta_t$ to explicitly satisfy a uniform high-probability bound for containing the true function values, instead just using a heuristic
- Assuming a known upper bound to the RKHS norm (this is very hard to calculate)

## Weakness 1: known Lipschitz constant

The authors correctly mention that there are many established methods for estimating Lipschitz constants from partial observations of a function. The authors vaguely argue that it is easier to get a good estimate of a Lipschitz constant compared to an RKHS norm, which I agree with. However, after repeatedly emphasizing how it is critical to have a super-high probability safety _guarantee_ and avoid hacky methods, the authors estimate the Lipschitz constant using a grid of evaluations, then multiply by 1.1 "just to be safe." This is very hypocritical: 1.1 is clearly another "heuristic constant" and this method could easily fail for suitably varying functions- think something like the [Weierstrass function](https://en.wikipedia.org/wiki/Weierstrass_function), which is "smooth" at a coarse level, but has low-level fluctuations.

Of course, Weierstrass-like functions are clearly pathological, and would not be anticipated in practice. Yet, if the authors assume some sort of  general smoothness, this feels like a qualitatively similar assumption to $f$ belonging in a smooth RKHS, no?

## Weakness 2: writing

This paper is, in my opinion, way too long for what it provides. It is clear that the authors have done a lot of reading and I appreciated thorough citations, but a lot of the discussion seemed tangential to the main points of the paper. Here are my thoughts section-by-section:

1. Good intro, but in the future if the paper is made shorter, maybe remove the huge "outline" paragraph (since it would be less necessary)
2. Intro to GPs makes sense. The introduction to RKHSs seems a bit too formal and long, given that RKHSs are not really used anywhere in the paper  (except for criticizing not knowing the RKHS norm). Section 2.3 seems very long and also not necessary: in the end all that is used is the bound in equation (7). Also, the sentence "_where the probability is with respect to the data-generating process, rather than to the target function f_" was confusing to me: are we not assuming that data is generated using $f$? Or are we assuming some fixed distribution over input values in addition to $f$?
3. This seemed like it should also belong in the "background", probably at the beginning, since it is the definition of the problem. The list O1-O3 seem duplicated with the intro and sections 5-6. The reader is already expecting this content, why list it again?
4. 4.1 has a lot of duplicate content with the intro. I think the reader gets the idea of why you would want to avoid "bad" function evaluations, no need to repeat this over and over again. 4.2 seems like a bunch of sentences that should just be elsewhere in the paper. The parts justifying the Lipschitz constant should probably be near Assumption 1. The parts mentioning sections 5-6 should probably just come after these sections.
5. This section basically says "you can't just set algorithm constants arbitrarily and expect safe behavior". I think this could be said much more succinctly. I don't think the experimental demonstration provided much value (of course heuristic values will not always work).
6. 6.1 seemed like it belongs in section 5.2, no? What does it have to do with Lipschitz constants? In 6.2, the algorithm could probably be stated first and then analyzed (currently Proposition 1 comes even before the statement of the algorithm). Also, I think it could be explained more intuitively (it is essentially just finding the regions which are guaranteed safe by the Lipschitz constant)
7. This algorithm seems very similar to the one in section 6. Maybe they can be presented together? The experiments were also very long. Consider moving some of it to the experiments?
8. Conclusion without much of a discussion of limitations

---

> ### Author Response · Authors · 2024-07-30
>
> Thank you very much for your detailed review and extensive comments.
>
> **Weakness 1 (known Lipschitz constant)**
> >However, after repeatedly emphasizing ... suitably varying functions
>
> We would like to stress that the above-mentioned method is used to determine a concrete Lipschitz bound in our experiments where we generate many synthetic test functions - we simply need a method to automatically determine a reasonable Lipschitz function for the large-scale numerical evaluation of the various algorithms (which curiously appears to have not been done in the literature before). Of course, the heuristic above should not be used in practice, exactly due to the potentially pathological behaviour you nicely outlined.
>
> >Yet, if the authors assume some sort of general smoothness, this feels like a qualitatively similar assumption to f belonging in a smooth RKHS, no?
>
> This is an excellent point, but there are important differences between the two assumptions.
> 1. In contrast to a (quantitative) bound on the RKHS norm, a (quantitative) Lipschitz bound has a very clear interpretation that can be linked to domain knowledge by practitioners - after all, it is just a bound on the rate of change of the target function.
> 2. In contrast to a (concrete) bound on the RKHS norm, (concrete) Lipschitz bounds are established in various application areas, most notably control.
> 3. Slightly more subtle, the RKHS norm depends on the chosen kernel, so even if the same function is contained in the RKHSs of two different kernels, a bound valid for one kernel might not be valid for another kernel. This is a particularly problematic issue in the context of unknown hyperparameters (and hyperparameter search). A Lipschitz-bound is considerably more "model-agnostic".
>
> **Requested changes**
> >The main change I would request is to add discussion contrasting the pros/cons of estimating RKHS norm bounds vs Lipschitz constant bounds.
>
> This is an excellent point, we will add such a discussion.
>
> >Also, acknowledging the failure mode of underexploring if the Lipschitz constant is high is also important to guide future users of the proposed algorithms.
>
> This is a good point, we will include an appropriate discussion.
>
> >I also highly highly suggest re-writing the paper to be much shorter (ideally < 10 pages). To do this, you can look at my notes above and think about re-structuring the paper by merging semi-duplicated sections and moving things to appendices.
>
> We strongly believe that many of the issues we tackle can be subtle and need appropriate discussion, which requires some space (after all, this was our main motivation to report our findings in a journal manuscript rather than a conference submission) in order to ensure precision and clarity (and reviewers ZXHe and Eqfg consider the manuscript to be well-written, which indicates that our writing approach worked). However, we agree that some parts of the manuscript can be made more concise, which we will implement. We would suggest to keep the structure as-is, but if you consider the length a severe problem, some parts of the article can be moved to an appendix (though we would argue that the current structure without an appendix aids the reader).
>
> >Clarify what input dimension was used for your experiments in sections 6-7 (this detail was not clear)
>
> We have checked the manuscript, the dimensionality is stated in Section 6 (1d) and 7 (1d for the first set of experiments, then 2d, 6d, 10d).
>
> >Also, small typos:
>
> Thank you very much, we will correct them.

---

> ### Author Response · Authors · 2024-07-30
> **Additional reply to the remarks on writing**
>
> **Weakness 2: writing**
> Thank you very much for your detailed remarks on the writing, to which we would like to reply in addition to our main answer from before.
>
> Section 1
> >maybe remove the huge "outline" paragraph
>
> Thank you for the suggestion, we will make the outline more concise.
>
> Section 2
> >The introduction to RKHSs seems a bit too formal and long, given that RKHSs are not really used anywhere in the paper (except for criticizing not knowing the RKHS norm)
>
> In order to keep the paper self-contained, we needed the material in Section 2.2, also for the discussion in Section 6.1, which in turn is important to motivate why a concrete upper bound on the RKHS norm is problematic (Section 6.1 is indeed rather lengthy, but since it is a subtle, but important issue, we decided to discuss it in depth). If the length is deemed a serious issue, parts of this discussion could be moved to an appendix.
>
> >Section 2.3 seems very long and also not necessary: in the end all that is used is the bound in equation (7).
>
> We agree, however, we felt necessary to explain why the bounds commonly used in SafeOpt-type algorithms are problematic, and why we choose the particular variant (7). Again, if the length is deemed a serious issue, parts of this discussion could be moved to an appendix.
>
> >Also, the sentence "where the probability is with respect to the data-generating process, rather than to the target function f" was confusing to me: are we not assuming that data is generated using f? Or are we assuming some fixed distribution over input values in addition to f?
>
> This is a good point, we will improve the wording. What is meant is that $f$ is a fixed function from which data is generated in some manner, involving potentially some randomness (usually via the additive measurement noise, but if a randomized algorithm is used, additional randomness enters).
>
> Section 3
> >This seemed like it should also belong in the "background", probably at the beginning, since it is the definition of the problem. The list O1-O3 seem duplicated with the intro and sections 5-6. The reader is already expecting this content, why list it again?
>
> On a high-level, there is indeed some duplication. However, because the technical issues from (O1) and (O2) rely on Section 2 and the detailed algorithmic description in Section 3, we decided to do it this way.
>
> Section 4
> >4.1 has a lot of duplicate content with the intro. I think the reader gets the idea of why you would want to avoid "bad" function evaluations, no need to repeat this over and over again.
>
> There is indeed some overlap, however, we wanted to accurately contextualize our work in the safe BO literature (in particular, making clear that our work is concerned with "hard safety" scenarios). We will make these paragraphs more concise / decrease the duplication.
>
> Section 5
> >This section basically says "you can't just set algorithm constants arbitrarily and expect safe behavior". I think this could be said much more succinctly. I don't think the experimental demonstration provided much value (of course heuristic values will not always work).
>
> While this is indeed the main message, in our opinion a detailed discussion with supporting experiments is necessary. However, some of the formulations can indeed be shortened, which we will implement.
>
> Section 6
> >6.1 seemed like it belongs in section 5.2, no? What does it have to do with Lipschitz constants?
>
> In Section 6.1 we provide a detailed discussion why the RKHS norm bound assumption is problematic in the context of SafeOpt-type algorithms, which in turn forms the central motivation for LoSBO. Of course, Section 6.1 would also fit in Section 5, but it appeared to us that this discussion fits better in Section 6 (since the core issue is going from an RKHS norm assumption to a Lipschitz bound).
>
> >In 6.2, the algorithm ... by the Lipschitz constant)
>
> Thank you very much for these suggestions, we will implement them.
>
> Section 7
> > Consider moving some of it to the experiments?
>
> The experiments are presented currently in a dedicated subsection. Do you mean putting the experiments in the appendix?

---

> > ### Comment · Reviewer_QF7s · 2024-07-31
> > **Thanks for the comments**
> >
> > Thank you for responding to the points raised in my review. I will read the revised manuscript once you post it. Regarding re-writing the paper, if you like its current form then I will not push you to do the re-writing. My concern is mainly that if it is too long then not many people will read it. Ultimately thought it is your paper so you should write it the way you think is best.

---

> > > ### Author Response · Authors · 2024-08-01
> > >
> > > Thank you very much for your answer and opinion on this issue. We agree that it could indeed be problematic, but it has to be balanced with ensuring a self-contained exposition and precise discussion of the (practical, not so much formal) safety issues, and the current format appeared to us as the best compromise.

---

### Review · Reviewer_ZXHe · 2024-07-18

**Summary Of Contributions:**

By deeply examining the practical implementation of a wide range of methods, specifically SafeOpt-type algorithms, the author questions three important aspects of these algorithms: 1) the use of heuristics instead of theoretically sound uncertainty bounds, 2) the assumption of an RKHS norm bound, and 3) the discrepancy between discrete input space in theory and continuous domain in moderately high dimensions in application. Overall, the motivations for addressing these three points are clear, and the proposed strategies for tackling these problems are novel.

**Audience:**

Yes

**Claims And Evidence:**

Yes

**Requested Changes:**

None

**Strengths And Weaknesses:**

Strengths:

1) The theoretical results are solid, and the paper is well-written.

2) The motivations are clearly articulated, with some supported by observations from experiments. Summaries in the main text provide valuable insights.

3) The paper addresses a key question regarding safety in safe Bayesian optimization.

Suggestions:
Overall, I think this work is good. Here are some suggestions (not necessary) that might be helpful.

1) I realize that some experiments in Section 5.1 may be sufficient to demonstrate that using heuristics in SafeOpt can be highly problematic. However, a theoretical analysis here would be more appreciated. Additionally, examining the underlying theory would provide more insight into why these heuristics are problematic and could lead to more interesting strategic approaches. I am trying to understand this part intuitively but find it difficult. Could you provide an interpretation of why such heuristics are problematic in an intuitive way?

2) Regarding the experimental design, I am curious about the performance of each strategy for each question. For example, how does the proposed solution compare to the heuristic solution on benchmarks, similar to the comparison of RealBetaSafeOpt versus LoSBO in Section 6. What about the remaining two solutions for the two questions, O(2) and O(3), in Section 3?

---

> ### Comment · Reviewer_QF7s · 2024-07-18
> **Intuitive explanation of heuristics**
>
> Hi reviewer ZXHe,
>
> The intuitive explanation of the heuristics is pretty simple: for a given domain, every $\delta$, there exists some value of $\beta$ such that the function values are contained within the confidence intervals of $\mu\pm\beta\sigma$ with probability $1-\delta$ (intuitively, smaller $\delta$ needs a larger $\beta$). The heuristic is that previous papers don't actually calculate the correct value of $\beta$ for a given value of $\delta$, they just make up a value, which might correspond to a very large $\delta$. Does that make sense?

---

> > ### Author Response · Authors · 2024-07-27
> >
> > Thank you very much for your answer, this is an excellent summary. In our answer below we provided some additional comments.

---

> ### Author Response · Authors · 2024-07-27
>
> Thank you very much for your detailed review and the positive evaluation of our work.
>
> Regarding Suggestion 1: Reviewer QF7s summed it up quite well. In SafeOpt-type algorithms, uncertainty bounds of the form (3) are needed, and there are indeed such bounds available (reviewed in Section 2.3), all of the form (4). However, to the best of our knowledge, these bounds have been used only in the theoretical analysis of SafeOpt-type algorithms, but not in the actual algorithms (or more precisely, their practical implementation and in corresponding experiments), probably because these bounds are usually stated in a form difficult to evaluate, and (beyond synthetic settings) the practical challenge to get an RKHS norm bound. Since some uncertainty bound is necessary in the actual algorithms, heuristics have been used, and the most simple one is to just set the $\beta_t$ to some constant.
>
> Intuitively, the scaling factors $\beta_t$ in (4) are conversion factors between the pointwise Bayesian uncertainty measures $\sigma_t(x)$ and frequentist (uniform in time and input) uncertainty bounds as in (3). It is clear that it can easily happen that an arbitrary scaling factor (say, $\beta_t \equiv 2$ as used very often) is not a valid conversion factor, so the bounds do not hold in the sense of (3), which in turn can lead to safety violations (as demonstrated in Section 5.1).
>
> Regarding Suggestion 2: Because SafeOpt, RealBetaSafeOpt, and LoSBO rely on a discrete-search space (so gridding is necessary for continuous domains), they cannot be used in practice for even moderate dimensional problems (the issue in (O3)), and hence we cannot compare them in experiments with LoS-GP-UCB. Furthermore, if an RKHS norm bound is available (as we argue this appears to be usually not the case in practice, cf. (O2)), then an algorithm like RealBetaSafeOpt should be used, and if heuristics are involved, then LoSBO is necessary (since we are in a hard safety scenario), hence we did not include (heuristic) SafeOpt-type algorithms in the experiments in Section 6.

---

### Author Response · Authors · 2024-08-01

Dear all,
thank you very much for the detailed reading, interesting questions and helpful comments. We have answered all the questions individually, and uploaded a revised version of the manuscript (a detailed description of the changes can be found in the revision section above).

The main changes are related to writing and exposition with the aim of making the manuscript more concise. A further shortening appears to be only possible by introducing an appendix and changing the structure of the manuscript.

Furthermore, we received several interesting questions below (to which we answered individually), but a detailed discussion (and corresponding experiments) appears to be out of the scope of the manuscript, though very relevant for follow-up work (in particular, in the context of safe BO with high dimensional instead of just moderately high dimensional inputs).

---

> ### Comment · Reviewer_QF7s · 2024-08-09
> **Appreciate revision**
>
> Thank you for posting the revision. I think the paper has improved and I submitted my recommendation for the paper to be accepted.
>
> That being said, I think shortening the paper would substantially increase the readership and help the paper have better long-term impact. I know it is a lot of work, but it might pay off in the long term. That's just my opinion though; I have already recommended acceptance so what you do from here is up to you.

---

### Decision · Action_Editor_tQAx · 2024-08-22

**Recommendation:** Accept with minor revision

**Comment:**

All reviewers agreed that the paper should be accepted in TMLR. See comments above regarding claims.

I do agree with Reviewer QF7s that this paper is quite a bit longer than it needs to be. I would strongly encourage the authors to shorten the paper by making the writing more concise throughout. The usual length of a TMLR paper is around 12 pages.

**Audience:**

This paper covers Bayesian optimization under a safety constraint, and proposes minor modifications to existing methods that appear to provide performance gains in certain settings.

**Claims And Evidence:**

Claims are generally well-supported. I do believe the paper is a bit misleading especially at the beginning but also throughout, in the sense that it emphasizes how past methods do not actually satisfy frequentist guarantees and how methods in this paper do. However, a major portion of how this paper gets around issues of past work is by simply "assuming away the problem". In particular
- for Real beta SafeOpt, the paper assumes the RKHS norm is known and then uses a previously known scaling
- for Lipschitz-only Safe Bayes Opt (and the UCB extension) the paper assumes hard bounds on noise magnitude and a known Lipschitz constant

The paper needs to be clearer at the beginning (and also throughout) about this limitation.